# ON INSTAHIDE, PHASE RETRIEVAL, AND SPARSE MATRIX FACTORIZATION

**Sitan Chen** *
MIT
sitanc@mit.edu.

**Xiaoxiao Li**
Princeton Universtiy
xiaoxiao.li@aya.yale.edu

**Zhao Song**
Columbia University, Princeton University / IAS
magic.linuxkde@gmail.com.

**Danyang Zhuo**
Duke University
danyang@cs.duke.edu.

## ABSTRACT

In this work, we examine the security of InstaHide, a scheme recently proposed by Huang et al. (2020b) for preserving the security of private datasets in the context of distributed learning. To generate a synthetic training example to be shared among the distributed learners, InstaHide takes a convex combination of private feature vectors and randomly flips the sign of each entry of the resulting vector with probability 1/2. A salient question is whether this scheme is secure in any provable sense, perhaps under a plausible complexity-theoretic assumption.

The answer to this turns out to be quite subtle and closely related to the average-case complexity of a multi-task, missing-data version of the classic problem of phase retrieval that is interesting in its own right. Motivated by this connection, under the standard distributional assumption that the public/private feature vectors are isotropic Gaussian, we design an algorithm that can actually recover a private vector using only the public vectors and a sequence of synthetic vectors generated by InstaHide.

## 1 INTRODUCTION

In distributed learning, where decentralized parties each possess some private local data and work together to train a global model, a central challenge is to ensure that the security of any individual party's local data is not compromised. Huang et al. (2020b) recently proposed an interesting approach called *InstaHide* for this problem. At a high level, InstaHide is a method for aggregating local data into synthetic data that can hopefully preserve the privacy of the local datasets and be used to train good models.

Informally, given a collection of *public feature vectors* (e.g. a publicly available dataset like ImageNet Deng et al. (2009)) and a collection of *private feature vectors* (e.g. the union of all of the private datasets among learners), InstaHide produces a synthetic feature vector as follows. Let integers $k_{\mathsf{pub}}, k_{\mathsf{priv}}$ be sparsity parameters.

1. Form a random convex combination of $k_{\mathsf{pub}}$ public and $k_{\mathsf{priv}}$ private vectors.

2. Multiply every coordinate of the resulting vector by an independent random sign in $\{\pm 1\}$, and define this to be the synthetic feature vector.

The hope is that by removing any sign information from the vector obtained in Step 1, Step 2 makes it difficult to discern which public and private vectors were selected in Step 1. Strikingly, Huang et al. (2020b) demonstrated on real-world datasets that if one trains a ResNet-18 or a NASNet on a

---

*This work was supported in part by NSF CAREER Award CCF-1453261, NSF Large CCF-1565235 and Ankur Moitra's ONR Young Investigator Award.

dataset consisting of synthetic vectors generated in this fashion, one can still get good *test accuracy* on the underlying private dataset for modest sparsity parameters (e.g. $k_{\mathsf{pub}} = k_{\mathsf{priv}} = 2$). [1]

The two outstanding *theoretical* challenges that InstaHide poses are understanding:

- **Utility**: What property, either of neural networks or of real-world distributions, lets one tolerate this kind of covariate shift between the synthetic and original datasets?

- **Security**: Can one rigorously formulate a refutable security claim for InstaHide, under a plausible average-case complexity-theoretic assumption?

In this paper we consider the latter question. One informal security claim implicit in Huang et al. (2020b) is that given a synthetic dataset of a certain size, no efficient algorithm can recover a private image to within a certain level of accuracy (see Problem 1 for a formal statement of this recovery question). On the one hand, it is a worthwhile topic of debate whether this is a satisfactory guarantee from a security standpoint. On the other, even this kind of claim is quite delicate to pin down formally, in part because it seems impossible for such a claim to hold for arbitrary private datasets.

**Known Attacks and the Importance of Distributional Assumptions**     If the private and public datasets consisted of natural images, for example, then attacks are known Jagielski (2020); Carlini et al. (2020). At a high level, the attack of Jagielski (2020) crucially leverages local Lipschitz-ness properties of natural images and shows that when $k_{\mathsf{priv}} + k_{\mathsf{pub}} = 2$, even a single synthetic image can reveal significant information. The very recent attack of Carlini et al. (2020), which was independent of the present work and appeared a month after this submission appeared online, is more sophisticated and bears interesting similarities to the algorithms we consider. We defer a detailed discussion of these similarities to Appendix A in the supplement.

While the original InstaHide paper Huang et al. (2020b) focused on image data, their general approach has the potential to be applicable to other forms of real-valued data, and it is an interesting mathematical question whether the above attacks remain viable. For instance, for distributions over private vectors where individual features are nearly independent, one cannot hope to leverage the kinds of local Lipschitz-ness properties that the attack of Jagielski (2020) exploits. Additionally, if the individual features are identically distributed, then it is information theoretically impossible to discern anything from just a single synthetic vector. For instance, if a synthetic vector $\widetilde{v}$ is given by the entrywise absolute value of $\frac{1}{2}v_1 + \frac{1}{2}v_2$ for private vectors $v_1, v_2$, then an equally plausible pair of private vectors generating $\widetilde{v}$ would be $v_1', v_2'$ given by swapping the $i$-th entry of $v_1$ with that of $v_2$ for any collection of indices $i \in [d]$. In other words, there are $2^d$ pairs of private vectors which are equally likely under the Gaussian measure and give rise to the exact same synthetic vector.

**Gaussian Images, and Our Results**     A natural candidate for probing whether such properties can make the problem of recovering private vectors more challenging is the case where the public and private vectors are sampled from the standard Gaussian distribution over $\mathbb{R}^d$. While this distribution does not capture datasets in the real world, it avoids some properties of distributions over natural images that might make InstaHide more vulnerable to attack and is thus a clean testbed for stress-testing candidate security claims for InstaHide. Furthermore, in light of known hardness results for certain learning problems over Gaussian space Diakonikolas et al. (2017); Bruna et al. (2020); Diakonikolas et al. (2020b); Goel et al. (2020a); Diakonikolas et al. (2020a); Klivans & Kothari (2014); Goel et al. (2020b); Bubeck et al. (2019); Regev & Vijayaraghavan (2017), one might hope that when the vectors are Gaussian, one could rigorously establish some lower bounds, e.g. on the size of the synthetic dataset (information-theoretic) and/or the runtime of the attacker (computational), perhaps under an average-case assumption, or in some restricted computational model like SQ.

Orthogonally, we note that the recovery task the attacker must solve appears to be an interesting inverse problem in its own right, namely a multi-task, missing-entry version of phase retrieval with an intriguing connection to sparse matrix factorization (see Section 2.2 and Section 3). The assumption of Gaussianity is a natural starting point for understanding the average-case complexity of this problem, and in this learning-theoretic context it is desirable to give algorithms with provable guarantees.

---

[1]We did not describe how the labels for the synthetic vectors are assigned, but this part of InstaHide will not be important for our theoretical results and we defer discussion of labels to Section 4.

Gaussianity is often a standard starting point for developing guarantees for such inverse problems Moitra & Valiant (2010); Netrapalli et al. (2013); Candes et al. (2015); Hardt & Price (2015); Zhong et al. (2017b;a); Li & Yuan (2017); Ge et al. (2018); Li & Liang (2018); Zhong et al. (2019); Chen et al. (2020); Kong et al. (2020); Diakonikolas et al. (2020b).

Our main result is to show that when the private and public data is Gaussian, we can use the synthetic and public vectors to recover a subset of the private vectors.

**Theorem 1.1** (Informal, see Theorem B.1). *If there are $n_{\mathsf{priv}}$ private vectors and $n_{\mathsf{pub}}$ public vectors, each of which is an i.i.d. draw from $\mathcal{N}(0, \mathsf{Id}_d)$, then as long as $d = \Omega(\mathrm{poly}(k_{\mathsf{pub}}, k_{\mathsf{priv}})$ $\log(n_{\mathsf{pub}} + n_{\mathsf{priv}}))$, there is some $m = o(n_{\mathsf{priv}}^{k_{\mathsf{priv}}})$ such that, given a sample of $m$ random synthetic vectors independently generated as above, one can* exactly recover $k_{\mathsf{priv}} + 2$ *private vectors in time $O(d(m^2 + n_{\mathsf{pub}}^2)) + \mathrm{poly}(n_{\mathsf{pub}})$ with probability 9/10 over the randomness of the private and public vectors and the randomness of the selection vectors.[2]*

We emphasize that we can take $m = o(n_{\mathsf{priv}}^{k_{\mathsf{priv}}})$, meaning we can achieve recovery even with access to a vanishing fraction of all possible combinations of private vectors among the synthetic vectors generated. For instance, when $k_{\mathsf{priv}} = 2$, we show that $m = O(n_{\mathsf{priv}}^{4/3})$ suffices (see Theorem B.1). See Remark B.2 for additional discussion.

Additionally, to ensure we are not working in an uninteresting setting where InstaHide has zero *utility*, we empirically verify that in the setting of Theorem 1.1, one can train on the synthetic vectors and get reasonable test accuracy on the original Gaussian dataset (see Section 4).

Qualitatively, the main takeaway of Theorem 1.1 is that to prove meaningful security guarantees for InstaHide, we must be careful about the properties we posit about the underlying distribution generating the public and private data, even in challenging settings where this data does not possess the nice properties of natural images that have made other attacks possible.

## 1.1 Connections and Extensions to Phase Retrieval

Our algorithm is based on connections and extensions to the classic problem of *phase retrieval*. At a high level, this can be thought of as the problem of linear regression where the signs of the linear responses are hidden. More formally, this is a setting where we get pairs $(x_1, y_1), ..., (x_N, y_N) \in \mathbb{C}^n \times \mathbb{R}$ for which there exists a vector $w \in \mathbb{C}^n$ satisfying $|\langle w, x_i \rangle| = y_i$ for all $i = 1, ..., N$, and the goal is to recover $w$. Without distributional assumptions on how $x_1, ..., x_N$ are generated, this problem is NP-hard Yi et al. (2014), and in the last decade, there has been a huge body of work, much of it coming from the machine learning community, on giving algorithms for recovering $w$ under the assumption that $x_1, ..., x_N$ are i.i.d. Gaussian, see e.g. Candes et al. (2013; 2015); Conca et al. (2015); Netrapalli et al. (2013).

To see the connection between InstaHide and phase retrieval, first imagine that InstaHide only works with public vectors (in the notation of Theorem 1.1, $n_{\mathsf{priv}} = k_{\mathsf{priv}} = 0$). Now, consider a synthetic vector $y \in \mathbb{R}^d$ generated by InstaHide, and let the vector $w \in \mathbb{R}^{n_{\mathsf{pub}}}$ be the one specifying the convex combination of public vectors that generated $y$. The basic observation is that for any feature $i \in [d]$, if $p_i \in \mathbb{R}^{n_{\mathsf{pub}}}$ is the vector consisting of $i$-th coordinates of all the public vectors, then $|\langle w, x_i \rangle| = y_i$.

In other words, if InstaHide only works with public vectors, then the problem of recovering which public vectors generated a given synthetic vector is formally *equivalent* to phase retrieval. In particular, if the public dataset is Gaussian, then we can leverage the existing algorithms for Gaussian phase retrieval. Huang et al. (2020b) already noted this connection but argued that if InstaHide also uses private vectors, the existing algorithms for phase retrieval fail. Indeed, consider the extreme case where InstaHide only works with *private* vectors (i.e. $n_{\mathsf{pub}} = 0$), so that the only information we have access to is the synthetic vector $(y_1, ..., y_d)$ generated by InstaHide. As noted above in the discussion about private distributions where the features are identically distributed, it is clearly information-theoretically impossible to recover anything about $w$ or the private dataset.

As we will see, the key workaround is to exploit the fact that InstaHide ultimately generates *multiple* synthetic vectors, each of which is defined by a *random* sparse convex combination of public/private

---

[2]See Problem 1 and Remark 2.7 for what exact recovery precisely means in this context.

vectors. And as we will make formal in Section 2.2, the right algorithmic question to study in this context can be thought of as a *multi-task*, *missing-data* version of phase retrieval (see Problem 2) that we believe to be of independent interest.

Lastly, we remark that in spite of this conceptual connection to phase retrieval, and apart from one component of our algorithm (see Section B.1) which draws upon existing techniques for phase retrieval, the most involved parts of our algorithm and its analysis utilize techniques that are quite different from the existing ones in the phase retrieval literature. We elaborate upon these techniques in Section 3.

## 2 TECHNICAL PRELIMINARIES

**Miscellaneous Notation** Given a subset $T$, let $\mathcal{C}_T^k$ denote the set of all subsets of $T$ of size exactly $k$. Given a vector $v \in \mathbb{R}^n$ and a subset $S \subseteq [n]$, let $[v]_S \in \mathbb{R}^{|S|}$ denote the restriction of $v$ to the coordinates indexed by $S$.

**Definition 2.1.** Given a Gaussian distribution $\mathcal{N}(0, \Sigma)$, let $\mathcal{N}^{\mathsf{fold}}(0, \Sigma)$ denote the *folded Gaussian distribution* defined as follows: to sample from $\mathcal{N}^{\mathsf{fold}}(0, \Sigma)$, sample $g \sim \mathcal{N}(0, \Sigma)$ and output $|g|$.

### 2.1 THE GENERATIVE MODEL

**Definition 2.2** (Image matrix notation). Let *image matrix* $\mathbf{X} \in \mathbb{R}^{d \times n}$ be a matrix whose columns consist of vectors $x_1, ..., x_n$ corresponding to $n$ images each with $d$ pixels taking values in $\mathbb{F}$.[3] It will also be convenient to refer to the rows of $\mathbf{X}$ as $p_1, ..., p_d \in \mathbb{R}^n$.

**Definition 2.3** (Public/private notation). Let $S \subset \{1, ..., n\}$ be some subset. We will refer to $S$ and $S^c \triangleq \{1, ..., n\} \backslash S$ as the set of *public* and *private* images respectively, and given a vector $w \in \mathbb{R}^n$, we will refer to $\mathrm{supp}(w) \cap S$ and $\mathrm{supp}(w) \cap S^c$ as the *public* and *private coordinates* of $w$ respectively.

**Definition 2.4** (Synthetic images). Given sparsity levels $k_{\mathsf{pub}} \leq |S|, k_{\mathsf{priv}} \leq |S^c|$, image matrix $\mathbf{X}$ and a *selection vector* $w \in \mathbb{R}^n$ for which $[w]_S$ and $[w]_{S^c}$ are $k_{\mathsf{pub}}$- and $k_{\mathsf{priv}}$-sparse respectively, the corresponding *synthetic image* is the vector

$$y^{\mathbf{X}, w} \triangleq |\mathbf{X}w|, \tag{1}$$

where $|\cdot|$ denotes entrywise absolute value. We say that $\mathbf{X}$ and a sequence of selection vectors $w_1, ..., w_m \in \mathbb{R}^n$ give rise to a *synthetic dataset* consisting of the images $\{y^{\mathbf{X}, w_1}, ..., y^{\mathbf{X}, w_m}\}$.

Note that instead of the entrywise absolute value of $\mathbf{X}w$, InstaHide in Huang et al. (2020b) randomly flips the sign of every entry of $\mathbf{X}w$, but these two operations are interchangeable in terms of information; it will be slightly more convenient to work with the former.

We will work with the following distributional assumption on the entries of $\mathbf{X}$:

**Definition 2.5** (Gaussian images). We say that $\mathbf{X}$ is a random *Gaussian image matrix* if its entries are sampled i.i.d. from $\mathcal{N}(0, 1)$.

We will also work with the following simple notion of "random convex combination" as our model for how the selection vectors $w_1, \ldots, w_m$ are generated:

**Definition 2.6** (Distribution over selection vectors). Let $\mathcal{D}$ be the distribution over selection vectors defined as follows. To sample once from $\mathcal{D}$, draw random subset $T_1 \subset S, T_2 \subseteq S^c$ of size $k_{\mathsf{pub}}$ and $k_{\mathsf{priv}}$ and output the unit vector whose $i$-th entry is $\frac{1}{\sqrt{k_{\mathsf{pub}}}}$ if $i \in T_1$, $\frac{1}{\sqrt{k_{\mathsf{priv}}}}$ if $i \in T_2$, and zero otherwise.[4]

The main algorithmic question we study is the following:

---

[3]We will often refer to public/private/synthetic feature vectors as *images*, and their coordinates as *pixels*, in keeping with the original applications of InstaHide to image datasets in Huang et al. (2020b)

[4]Note that any such vector does not specify a convex combination, but this choice of normalization is just to make some of the analysis later on somewhat cleaner, and our results would still hold if we chose the vectors in the support of $\mathcal{D}$ to have entries summing to 1.

---

*Problem* 1 (Private (exact) image recovery). Let $\mathbf{X} \in \mathbb{R}^{d \times n}$ be a Gaussian image matrix. Given access to the public images $\{x_s\}_{s \in S}$ and the synthetic dataset $\{y^{\mathbf{X}, w_1}, \ldots, y^{\mathbf{X}, w_m}\}$, where $w_1, \ldots, w_m \sim \mathcal{D}$ are unknown selection vectors, output a vector $x \in \mathbb{R}^d$ for which there exists private image $x_s$ (where $s \in S^c$) satisfying $|x_i| = |(x_s)_i|$ for all $i \in [d]$.

---

*Remark* 2.7. Note that it is information-theoretically impossible to guarantee that $x_i = (x_s)_i$. This is because the distribution over $\mathbf{X}$ and the distribution over matrices given by sampling $\mathbf{X}$ and multiplying every private image by -1 are both Gaussian. And if the selection vectors $w_1, \ldots, w_m$ generated the synthetic images in the former case, then the selection vectors $w'_1 \ldots, w'_m$, where $w'_j$ is obtained by multiplying the private coordinates of $w_j$ by -1, would generate the exact same synthetic images.

## 2.2 MULTI-TASK PHASE RETRIEVAL WITH MISSING DATA

In this section we make formal the discussion in Section 1.1 and situate it in the notation above. First consider a synthetic dataset consisting of a single image $y \triangleq y^{\mathbf{X}, w}$, where $w$ is arbitrary and $\mathbf{X}$ is a random Gaussian image. From Eq. (1) we know that

$$|\langle w, p_j \rangle| = y_j \ \forall \ j \in [d].$$

If $S = \{1, \ldots, n\}$, then the problem of recovering selection vector $w$ from synthetic dataset $\{y\}$ is merely that of recovering $w$ from pairs $(p_j, y_j)$, and this is exactly the problem of *phase retrieval* over Gaussians. More precisely, because $w$ is assumed to be sparse, this is the problem of *sparse phase retrieval* over Gaussians.

If $S \subsetneq \{1, \ldots, n\}$, then it's clearly impossible to recover the private coordinates of $w$ from $y^{\mathbf{X}, w}$ alone. But it may still be possible to recover the public coordinates: formally, we can hope to recover $[w]_S$ given pairs $([p_j]_S, y_j)$, where the $p_j$'s are sampled independently from $\mathcal{N}(0, \mathsf{Id}_n)$. This can be thought of as a *missing-data* version of sparse phase retrieval where some known subset of the coordinates of the inputs, those indexed by $S^c$, are unobserved.

But recall our ultimate goal is to say something about the *private images*. It turns out that because we actually observe multiple synthetic images, corresponding to multiple vectors $w$, it becomes possible to recover $x_s$ for some $s \in S^c$ (even in the extreme case where $S = \emptyset$!). This corresponds to the following inverse problem which is *formally equivalent* to Problem 1, but phrased in a self-contained way which may be of independent interest.

---

*Problem* 2 (Multi-task phase retrieval with missing data). Let $S \subsetneq [n]$ and $S^c = [n] \backslash S$. Let $\mathbf{X} \in \mathbb{R}^{d \times n}$ be a matrix whose entries are i.i.d. draws from $\mathcal{N}(0, 1)$, with rows denoted by $p_1, \ldots, p_d$ and columns denoted by $x_1, \ldots, x_n$. Let $w_1, \ldots, w_m \sim \mathcal{D}$.

For every $j \in [d]$, we get a tuple $\left( [p_j]_S, y_j^{(1)}, \ldots, y_j^{(m)} \right)$ satisfying

$$|\langle w_i, p_j \rangle| = y_j^{(i)} \ \forall \ i \in [m], j \in [d].$$

Using just these, output $x \in \mathbb{R}^d$ such that for some $s \in S^c$, $|x_i| = |(x_s)_i|$ for all $i \in [d]$.

---

## 3 PROOF OVERVIEW

At a high level, our algorithm has three components:

1. Learn the public coordinates of all the selection vectors $w_1, \ldots, w_m$ used to generate the synthetic dataset.

2. Recover the $m \times m$ rescaled Gram matrix $\mathbf{M}$ whose $(i, j)$-th entry is $k \cdot \langle w_i, w_j \rangle$.

3. Use $\mathbf{M}$ and the synthetic dataset to recover a private image.

Step 1 draws upon techniques in Gaussian phase retrieval, while Step 2 follows by leveraging the correspondence between the covariance matrix of a Gaussian and the covariance matrix of its corresponding *folded Gaussian* (see Definition 2.1). Step 3 is the trickiest part and calls for leveraging delicate properties of the distribution $\mathcal{D}$ over selection vectors.

**Learning the Public Coordinates of Any Selection Vector**   We begin by describing how to carry out Step 1 above. First consider the case where $S = \{1, \ldots, n\}$, that is, where every image is public. Recall from the discussion in Section 1.1 and 2.2 that in this case, the question of recovering $w$ from synthetic image $y^{\mathbf{X},w}$ is equivalent to Gaussian phase retrieval. One way to get a reasonable approximation to $w$ is to consider the $n \times n$ matrix

$$\mathbf{N} \triangleq \mathop{\mathbf{E}}_{p,y}[y^2 \cdot (pp^\top - \mathsf{Id})], \qquad p \sim \mathcal{N}(0, \mathsf{Id}_n) \text{ and } y = |\langle w, p\rangle|.$$

It is a standard calculation (see Lemma B.3) to show that $\mathbf{N}$ is a rank-one matrix proportional to $ww^\top$. And as every one of $p_1, \ldots, p_d$ is an independent sample from $\mathcal{N}(0, \mathsf{Id})$, and $y_i^{\mathbf{X},w}$ satisfies $\langle w, p_i \rangle = y_i^{\mathbf{X},w}$ for every pixel $i \in [d]$, one can approximate $\mathbf{N}$ with the matrix

$$\widehat{\mathbf{N}} \triangleq \frac{1}{d} \sum_{i=1}^{d} \left( y_i^{\mathbf{X},w} \right)^2 \cdot (p_i p_i^\top - \mathsf{Id}).$$

This is the basis for the spectral initialization procedure that is present in many works on Gaussian phase retrieval, see e.g. Candes et al. (2015); Netrapalli et al. (2013). $\widehat{\mathbf{N}}$ will not be a sufficiently good spectral approximation to $\mathbf{N}$ when $d \ll n$, so instead we use a standard post-processing step based on the canonical SDP for sparse PCA (see (2)). Instead of taking the top eigenvector of $\widehat{\mathbf{N}}$, we can take the top eigenvector of the SDP solution and argue that as long as $d = \widetilde{\Omega}(\text{poly}(k_{\mathsf{pub}}) \log n)$, this will be sufficiently close to $w$ that we can exactly recover $\text{supp}(w)$.

Now what happens when $S \subsetneq \{1, \ldots, n\}$? Interestingly, if one simply modifies the definition of $\mathbf{N}$ to be $\mathbf{E}_{p,y}[y^2 \cdot ([p]_S [p]_S^\top - \mathsf{Id})]$ and defines the corresponding empirical analogue $\widehat{\mathbf{N}}$ formed from the pairs $\{([p_i]_S, y_i^{\mathbf{X},w})\}_{i \in [d]}$, one can still argue (see Lemma B.3) that the $\mathbf{N}$ is a rank-1 $|S| \times |S|$ matrix proportional to $[w]_S [w]_S^\top$ and that the top eigenvector of the solution to a suitable SDP formed from $\widehat{\mathbf{N}}$ will be close to $w$ (see Lemma B.4).

**Recovering the Gram Matrix via Folded Gaussians**   As we noted earlier, it is information-theoretically impossible to recover $[w_i]_{S^c}$ for any $i \in [m]$ given only $y^{\mathbf{X},w_i}$ and $[w_i]_S$, but we now show it's possible to recover the inner products $\langle [w_i]_{S^c}, [w_j]_{S^c} \rangle$ for any $i, j \in [m]$. For the remainder of the overview, we will work in the extreme case where $S^c = \{1, ..., n\}$, though it is not hard (see Section B.7) to combine the algorithms we are about to discuss with the algorithm for recovering the public coordinates to handle the case of general $S$. For brevity, let $k \triangleq k_{\mathsf{priv}}$.

First note that the $m \times d$ matrix whose rows consist of $y^{\mathbf{X},w_1}, ..., y^{\mathbf{X},w_m}$ can be written as

$$\mathbf{Y} \triangleq \begin{pmatrix} |\langle p_1, w_1\rangle| & \cdots & |\langle p_d, w_1\rangle| \\ \vdots & \ddots & \vdots \\ |\langle p_1, w_m\rangle| & \cdots & |\langle p_d, w_m\rangle| \end{pmatrix}.$$

Observe that without absolute values, each column would be an independent draw from the $m$-variate Gaussian $\mathcal{N}(0, \mathbf{M})$, where $\mathbf{M}$ is the Gram matrix defined above. Instead, with the absolute values, each column of $\mathbf{Y}$ is actually an independent draw from the *folded Gaussian* $\mathcal{N}^{\mathsf{fold}}(0, \mathbf{M})$ (Definition 2.1). The key point is that the covariance of $\mathcal{N}^{\mathsf{fold}}(0, \mathbf{M})$ can be directly related to $\mathbf{M}$ (see Corollary B.6), so by estimating the covariance of the folded Gaussian $\mathcal{N}^{\mathsf{fold}}(0, \mathbf{M})$ using the columns of $\mathbf{Y}$, we can obtain a good enough approximation $\widetilde{\mathbf{M}}$ to $\mathbf{M}$ that we can simply round every entry of $\widetilde{\mathbf{M}}$ so that the rounded matrix exactly equals $\mathbf{M}$. Furthermore, we only need to *entrywise* approximate the covariance of $\mathcal{N}^{\mathsf{fold}}(0, \mathbf{M})$ for all this to work, which is why it suffices for $d$ to grow *logarithmically* in $m$.

**Discerning Structure From the Gram Matrix**   By this point, we have access to the Gram matrix $\mathbf{M}$. Equivalently, we now know for any $i, j \in [m]$ whether $\text{supp}(w_i) \cap \text{supp}(w_j) \neq \emptyset$, that is, for any pair of synthetic images, we can tell whether the set of private images generating one of them overlaps with the set generating the other. Note that $\mathbf{M} = k \cdot \mathbf{W}\mathbf{W}^\top$, where $\mathbf{W}$ is the matrix whose $i$-th row is $w_i$, so if we could factorize $\mathbf{M}$ in this way, we would be able to recover which private vectors generated each synthetic vector. Of course, this kind of factorization problem, even if we constrain the factors to be row-sparse like $\mathbf{W}$, has multiple solutions. One reason is that any

permutation of the columns of $\mathbf{W}$ would also be a viable solution, but this is not really an issue because the ordering of the private images is not identifiable to begin with.

A more serious issue is that if $m$ is too small, there might be row-sparse factorizations of $\mathbf{M}$ which appear to be valid, but which we could definitively rule out upon sampling more synthetic vectors. For instance, suppose the first $k+1$ selections vectors all satisfied $|\mathrm{supp}(w_j) \cap \mathrm{supp}(w_{j'})| = k-1$. Ignoring the fact that this is highly unlikely, in such a scenario it is impossible to distinguish between the case where the corresponding synthetic images all have the same $k-1$ private images in common, and the case where there is a group $T \subseteq [n]$ of $k+1$ private images such that each of these synthetic images is comprised of a subset of $T$ of size $k$. But if we then sampled a new selection vector $w_{k+2}$ for which $|\mathrm{supp}(w_j) \cap \mathrm{supp}(w_{k+2})| = 1$ for all $j \in [k+1]$, we could rule out the latter.

This is indicative of a more general issue, namely that one cannot always recover the identity of a collection of subsets (even up to relabeling) if one only knows the sizes of their pairwise intersections!

This leads to the following natural combinatorial question. What families of sets are *uniquely identified* (up to trivial ambiguities) by the sizes of the pairwise intersections? One answer to this question, as we show, is the family of all subsets of $\{1, ..., k+2\}$ of size $k$ (see Lemma B.11). This leads us to the following definition:

**Definition 3.1** (Floral Submatrices). A $\binom{k+2}{k} \times \binom{k+2}{k}$ matrix $\mathbf{H}$ is *floral* if the following holds. Fix some lexicographic ordering on $\mathcal{C}_{[k+2]}^k$ and index $\mathbf{H}$ according to this ordering. There is some permutation matrix $\mathbf{\Pi}$ for which the matrix $\mathbf{H}' \triangleq \mathbf{\Pi}^\top \mathbf{H} \mathbf{\Pi}$ satisfies that for every pair of $S, S' \in \mathcal{C}_{[k+2]}^k$, $\mathbf{H}'_{S,S'} = |S \cap S'|$. See Example B.18 in the supplement.

The upshot is that if we can identify a floral submatrix of $\mathbf{M}$, then we know for certain that the subsets of private images picked by those selection vectors comprise all size-$k$ subsets of some subset of $[n]$ of size $k+2$. In summary, using the pairwise intersection size information provided by the Gram matrix $\mathbf{M}$, we can pinpoint collections of selection vectors which share a nontrivial amount of common structure.

**Learning a Private Image With a Floral Submatrix**  What can we do with this common structure in a floral submatrix? Let $t = \binom{k+2}{k}$. Given that the selection vectors $w_{i_1}, ..., w_{i_t}$ corresponding to the rows of the floral submatrix only involve $k+2$ different private images altogether, and there are $t > k+2$ constraints of the form $|\langle w_{i_j}, p_\ell \rangle| = y_\ell^{\mathbf{X}, w_{i_j}}$ for any pixel $\ell \in [d]$, we can hope that for each pixel, we can uniquely recover the $k+2$ private images from solving this system of equalities, where the unknowns are the values of the $k+2$ private images at that particular pixel. *A priori*, the fact that the number of constraints in this system exceeds the number of unknowns does not immediately guarantee that this system has a unique solution up to multiplying the solution uniformly by $-1$. Here we exploit the fact that $\mathbf{X}$ is Gaussian to show however that this is the case almost surely (Lemma B.10). Finally, note that this system can be solved in time $\exp(O(k^2))$ by simply enumerating over $2^t$ sign patterns. We conclude that if we could find a floral submatrix, then we would find not just one, but in fact $k+2$ private images!

**Existence of Floral Submatrix, and How to Find It**  It remains to understand how big $m$ has to be before we can guarantee the existence of a floral submatrix inside the Gram matrix $\mathbf{M}$ with high probability. Obviously if $m$ were big enough that with high probability we see *every possible* synthetic image that could arise from a selection vector $w$ in the support of $\mathcal{D}$, then $\mathbf{M}$ will contain many floral submatrices. One surprising part of our result is that we can ensure the existence of a floral submatrix when $m$ is much smaller. Our proof of this is quite technical, but at a high level it is based on the second moment method (see Lemma B.12).

The final question is: provided a floral submatrix of $\mathbf{M}$ exists, *how do we find it*? Note that naively, we could always brute-force over all $\binom{m}{O(k^2)} \leq n^{O(k^3)}$ principal submatrices with exactly $\binom{k+2}{k}$ rows/columns, and for each such principal submatrix we can check in $\exp(\widetilde{O}(k))$ time whether it is floral.

Surprisingly, we give an algorithm that can identify a floral submatrix of $\mathbf{M}$ in time dominated by the time it takes to write down the entries of the Gram matrix $\mathbf{M}$. Note that an off-the-shelf algorithm for

subgraph isomorphism would not suffice as the size of the submatrix in question is $O(k^2)$, and furthermore such an algorithm would need to work for *weighted graphs*. Instead, our approach is to use the constructive nature of the proof in Lemma B.11, that the family of all subsets of $\{1, ..., k+2\}$ of size $k$ is uniquely identified by the sizes of the pairwise intersections. By algorithmizing this proof, we give an efficient procedure for finding a floral submatrix, see Algorithm 3 and Lemma B.17. An important fact we use is that if we restrict our attention to the entries of $\mathbf{M}$ equal to $k-1$ or $k-2$, this corresponds to a graph over the selection vectors which is sparse with high probability.

We defer the formal specification and analysis of our algorithm to the supplement.

## 4 EXPERIMENTS

We describe an experiment demonstrating the *utility* of InstaHide for Gaussian images and comparing to the utility of another data augmentation scheme, MixUp Zhang et al. (2018). We also informally report on our implementation of LEARNPUBLIC and its empirical efficacy.

### 4.1 CHOICE OF ARCHITECTURE AND PARAMETERS

As our empirical results are purely for proof-of-concept, we work with a fairly basic neural network architecture. We use a 4-layer neural network as a binary classifier,

$$y = \arg\max(\mathrm{softmax}(W_4\sigma(W_3\sigma(W_2(\sigma(W_1x + b_1)) + b2) + b3) + b4)),$$

where $x \in \mathbb{R}^{10}$, $W_1 \in \mathbb{R}^{100\times 10}$, $W_2 \in \mathbb{R}^{100\times 100}$, $W_3 \in \mathbb{R}^{100\times 100}$, $W_4 \in \mathbb{R}^{2\times 100}$, $b_1 \in \mathbb{R}^{100}$, $b_2 \in \mathbb{R}^{100}$, $b_3 \in \mathbb{R}^{100}$, $b_4 \in \mathbb{R}^{100}$. We initialize the entries of each $W_l$ and $b_l$ to be i.i.d. draws from $\mathcal{N}(u_l, 1)$, where $u_l$ is sampled from $\mathcal{N}(0, \alpha)$ at the outset. We train the neural network for 100 epochs with cross-entropy loss and SGD optimizer with a learning rate of $0.01$. We do not need to distinguish between public and private images in our experiments, so let $k_{\mathsf{priv}} = 0$ and $k_{\mathsf{pub}} = k$ for $k \in \{1, 2, 3, 6\}$, and for each choice of $k$, we use random $k$-sparse selection vectors whose nonzero entries equal $1/k$. In all of our experiments, we separate our original image data (before generating synthetic data) into two categories: 80% training data and 20% test data. We train on synthetic images generated by MixUp or InstaHide using the training data, and measure the "test accuracy" on the training data and the test data separately. We provide more choices of $k$ in Appendix C.

### 4.2 GAUSSIAN DATA

**Settings** We considered binary classification on Gaussian data. We generated random images $x_1, \ldots, x_{1000} \in \mathbb{R}^{10}$ from $\mathcal{N}(0, \mathsf{Id})$ and a random vector $v \in \mathbb{R}^{10} \sim \mathcal{N}(0, \mathsf{Id})$. We then ranked all the Gaussian images based on $\sum_i |x_i v_i|$ and labeled the largest half as '1' and the rest as '0'. The point of choosing this labeling function is that it would assign the same label to any $x, x'$ which agree entrywise in magnitudes. Given a synthetic image generated via MixUp or InstaHide using selection vector $w$, we assigned it the label which is the convex combination of the one-hot encodings of the labels of the original images indexed by $w$.

**Results** We compare training and test loss over epochs when training on a synthetic dataset generated by either MixUp or Instahide, as shown in Figure 1. We use the convention in this paper of defining synthetic images under InstaHide to have all nonnegative entries (rather than imposing random sign flips), though we explore in the supplement how random sign flips can affect learnability. Compared to training on MixUp, InstaHide results on lower model performance (accuracy). As we expected, when we increase the $k$, both MixUp and InstaHide suffer from accuracy loss. Instahide dropped by $\sim 10\%$ accuracy when $k = 6$ compared to classical training $k = 1$, while MixUp dropped by $\sim 5\%$.

### 4.3 IMPLEMENTATION OF LEARNPUBLIC

We implemented LEARNPUBLIC for $k_{\mathsf{priv}} = 2$ and $n \in \{2000, 5000, 7500, 10000\}$. For $k_{\mathsf{pub}} = 2, 4, 6$, we respectively chose $d = 1000, 1800, 2400$. In particular, our choice of $d$ is meant to work essentially for any choice of $n$ (modulo the logarithmic dependence which does not noticeably

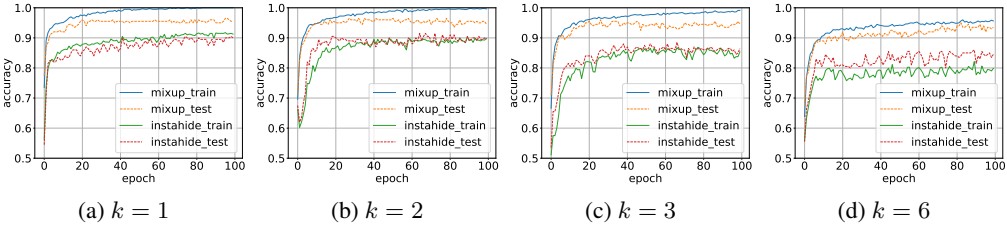

(a) $k = 1$      (b) $k = 2$      (c) $k = 3$      (d) $k = 6$

Figure 1: Comparing MixUp and Instahide training on Gaussian dataset with different $k$.

manifest in this regime). One heuristic modification that we made to LEARNPUBLIC was to use a *diagonal thresholding* approach from Cai et al. (2016) in place of solving the SDP in (2): namely for every $j \in [n]$ we computed the quantity $\frac{1}{d} \sum_{i=1}^{d} y_i^2 \cdot (x_i)_j^2$, zeroed out all but the principal submatrix of $\widetilde{\mathbf{M}}$ indexed by the top 25 such $j$, and computed the top eigenvector of the resulting matrix. For each parameter setting we found that, as expected, we were able to recover an average of at least 90% of the support. As this experiment was primarily to demonstrate that $d$ can be much less than $n$, we did not explore further optimizations to the algorithm.

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

## A    DISCUSSION OF OTHER ATTACKS

**Attack of Jagielski (2020)**    It has been pointed out Jagielski (2020) that for $k_{\mathsf{priv}} = 2, k_{\mathsf{pub}} = 0$, given a single synthetic image one can discern large regions of the constituent private images simply by taking the entrywise absolute value of the synthetic image. The reason is the pixel values of a natural image are mostly continuous, i.e. nearby pixels typically have similar values, so the entrywise absolute value of the InstaHide image should be similarly continuous. That said, natural images have enough discontinuities that this breaks down if one mixes more than just two images, and as discussed above, this attack is not applicable when the individual private features are i.i.d. like in our setting.

**Attack of Carlini et al. (2020)**    A month after this submission, Carlini et al. Carlini et al. (2020) independently gave an attack breaking the InstaHide challenge originally released by the authors of Huang et al. (2020b). In that challenge, the public dataset was ImageNet, the private dataset consisted of $n_{\mathsf{priv}} = 100$ natural images, and $k_{\mathsf{priv}} = 2$, $k_{\mathsf{pub}} = 4$, $m = 5000$. They were able to produce a visually similar copy of each private image.

Most of their work goes towards recovering which private images contributed to each synthetic image. Their first step is to train a neural network on the public dataset to compute a *similarity matrix* with rows and columns indexed by the synthetic dataset, such that the $(i, j)$-th entry approximates the indicator for whether the pair of private images that are part of synthetic image $i$ overlaps with the pair that is part of synthetic image $j$. Ignoring the rare event that two private images contribute to two distinct synthetic images, and ignoring the fact that the accuracy of the neural network for estimating similarity is not perfect, this similarity matrix is precisely our Gram matrix in the $k_{\mathsf{priv}} = 2$ case.

The bulk of Carlini et al.'s work Carlini et al. (2020) is focused on giving a heuristic for factorizing this Gram matrix. They do so essentially by greedily decomposing the graph with adjacency matrix given by the Gram matrix into $n_{\mathsf{priv}}$ cliques (plus some k-means post-processing) and regarding each clique as consisting of synthetic images which share a private image in common. They then construct an $m \times n_{\mathsf{priv}}$ bipartite graph as follows: for every synthetic image index $i$ and every private image index $j$, connect $i$ to $j$ if for four randomly chosen elements $i_1, ..., i_4 \in [m]$ of the $j$-th clique, the $(i, i_\ell)$-th entries of the Gram matrix are nonzero. Finally, they compute a min-cost max-flow on this instance to assign every synthetic image to exactly $k_{\mathsf{priv}} = 2$ private images.

It then remains to handle the contribution from the public images. Their approach is quite different from our sparse PCA-based scheme. At a high level, they simply pretend the contribution from the public images is mean-zero noise and set up a nonconvex least-squares problem to solve for the values of the constituent private images.

**Comparison to Our Generative Model**    Before we compare our algorithmic approach to that of Carlini et al. (2020), we mention an important difference between the setting of the InstaHide challenge and the one studied in this work, namely the way in which the random subset of public/private images that get combined into a synthetic image is sampled. In our case, for each synthetic image, the subset is chosen independently and uniformly at random from the collection of all subsets consisting of $k_{\mathsf{priv}}$ private images and $k_{\mathsf{pub}}$ public images. For the InstaHide challenge, batches of $n_{\mathsf{priv}}$ synthetic images get sampled one at a time via the following process: for a given batch, sample two random permutations $\pi_1, \pi_2$ on $n_{\mathsf{priv}}$ elements and let the $t$-th synthetic image in this batch be given by combining the private images indexed by $\pi_1(t)$ and $\pi_2(t)$. Note that this process ensures that every private image appears *exactly* $2m/n_{\mathsf{priv}}$ times, barring the rare event that $\pi_1(t) = \pi_2(t)$ for some $t$ in some batch. It remains to be seen to what extent the attack of Carlini et al. (2020) degrades in the absence of this sort of regularity property in our setting.

**Comparison to Our Attack**    The main commonality between our approach and that of Carlini et al. (2020) is to identify the question of extracting private information from the Gram matrix as the central algorithmic challenge.

How we compute this Gram matrix differs. We use the relationship between covariance of a folded Gaussian and covariance of a Gaussian, while Carlini et al. (2020) use the public dataset to train a neural network on public data to approximate the Gram matrix.

How we use this matrix also differs significantly. We do not produce a candidate factorization but instead pinpoint a collection of synthetic images such that we can provably ascertain that each one comprises $k_{\mathsf{priv}}$ private images from the same set of $k_{\mathsf{priv}} + 2$ private images. This allows us to set up an appropriate piecewise linear system of size $O(k_{\mathsf{priv}})$ with a provably unique solution and solve for the $k_{\mathsf{priv}} + 2$ private images.

An exciting future direction is to understand how well the heuristic in Carlini et al. (2020) scales with $k_{\mathsf{priv}}$. Independent of the connection to InstaHide, it would be very interesting from a theoretical standpoint if one could show that their heuristic provably solves the multi-task phase retrieval problem defined in Problem 2 in time scaling only polynomially with $k_{\mathsf{priv}}$ (i.e. the sparsity of the vectors $w_1, \ldots, w_m$ in the notation of Problem 2).

## B  RECOVERING PRIVATE IMAGES FROM A GAUSSIAN DATASET

In this section we prove our main algorithmic result:

**Theorem B.1** (Main). *Let $S \subsetneq [n]$, and let $n_{\mathsf{pub}} = |S|$ and $n_{\mathsf{priv}} = |S^c|$. Let $k = k_{\mathsf{pub}} + k_{\mathsf{priv}}$. If $d \geq \Omega(\mathrm{poly}(k_{\mathsf{pub}}, k_{\mathsf{priv}}) \cdot \log(n_{\mathsf{pub}} + n_{\mathsf{priv}}))$ and $m \geq \Omega\left(n_{\mathsf{priv}}^{k_{\mathsf{priv}} - \frac{2}{k_{\mathsf{priv}}+1}} k^{\mathrm{poly}(k_{\mathsf{priv}})}\right)$, then with high probability over $\mathbf{X}$ and the sequence of randomly chosen selection vectors $w_1, \ldots, w_m \sim \mathcal{D}$, there is an algorithm which takes as input the synthetic dataset $\{y^{\mathbf{X}, w_i}\}_{i \in [m]}$ and the columns of $\mathbf{X}$ indexed by $S$, and outputs $k_{\mathsf{priv}} + 2$ distinct images $\widetilde{x}_1, \ldots, \widetilde{x}_{k_{\mathsf{priv}}+2}$ for which there exist $k_{\mathsf{priv}} + 2$ distinct private images $x_{i_1}, \ldots, x_{i_{k_{\mathsf{priv}}+2}}$ satisfying $|\widetilde{x}_j| = |x_{i_j}|$ for all $j \in [k_{\mathsf{priv}} + 2]$. Furthermore, the algorithm runs in time*

$$O(dm^2 + dn_{\mathsf{pub}}^2 + n_{\mathsf{pub}}^{2\omega+1}).$$

*where $\omega \approx 2.373$ is the exponent of matrix multiplication.*

*Remark* B.2. Here we give some interpretation to the quantitative guarantees of Theorem B.1:

- The number of pixels $d$ only needs to depend logarithmically on the number of public/private images and polynomially in the sparsity $k_{\mathsf{pub}}, k_{\mathsf{priv}}$, which will be some small positive integer (e.g. $k_{\mathsf{pub}} + k_{\mathsf{priv}} = 4$ or $8$ in Huang et al. (2020a), $k_{\mathsf{pub}} + k_{\mathsf{priv}} = 4$ or $6$ in Huang et al. (2020b) and $k_{\mathsf{pub}} + k_{\mathsf{priv}} = 2$ in the implementation of MixUp in Zhang et al. (2018)), so the regime in which Theorem B.1 applies is quite realistic.

- Note that we can achieve recovery even when $m = o(n_{\mathsf{priv}}^{k_{\mathsf{priv}}})$. The reason this is significant is that as soon as $m = \Omega(n_{\mathsf{priv}}^{k_{\mathsf{priv}}})$, all possible combinations of $k$ private images are used. While it is still not immediately clear how to recover private images once this has happened, we regard the fact that we can do so well before this point to be one of the most interesting aspects of our result. Finally, we remark that the runtime is largely dominated by the $O(m^2)$ term coming from forming an $m \times m$ matrix whose $(i, j)$-th entry turns out to equal $\langle w_i, w_j \rangle$ for all $i, j \in [m]$. In fact, naive implementations of the most sophisticated part of our algorithm (see Sections B.3, B.4, B.5, and B.6) require time $\omega(m^2)$, and getting these parts of the algorithm to run in $O(m^2)$ time turns out to be quite subtle.

### B.1  LEARNING THE PUBLIC COORDINATES VIA GAUSSIAN PHASE RETRIEVAL

In this section we give a procedure which, given any synthetic image $y^{\mathbf{X}, w}$, recovers the entire support of $[w]_S$. The algorithm is inspired by existing algorithms for sparse phase retrieval, with the catch that we need to handle the fact that we only get to observe the public subset of coordinates of any of the vectors $p_j$. Our algorithm, LEARNPUBLIC is given in Algorithm 1 below.

We first show that the population version of the matrix $\widetilde{\mathbf{M}}$ formed in Step 1 is a rank-1 projector whose top eigenvector is in the direction of $[w]_S$.

**Lemma B.3.** *Let $w$ be a unit vector. Let $\widetilde{\mathbf{M}} \in \mathbb{R}^{n \times n}$ be defined as*

$$\widetilde{\mathbf{M}} \triangleq \frac{1}{d} \sum_{j=1}^{d} (y_j^2 - 1) \cdot \left([p_j]_S \cdot [p_j]_S^\top - \mathsf{Id}\right)$$

---

**Algorithm 1:** LEARNPUBLIC($\{([p_j]_S, y_j)\}_{j \in [d]}$)

---

**Input:** Samples $([p_1]_S, y_1), ..., ([p_d]_S, y_d)$
**Output:** $\mathrm{supp}([w]_S)$ with probability at least $1 - \delta$, provided $d \geq \mathrm{poly}(k_{\mathsf{pub}})/\log(n/\delta)$

1 Form the matrix $\widetilde{\mathbf{M}} \triangleq \frac{1}{d} \sum_{j=1}^{d} (y_j^2 - 1) \cdot ([p_j]_S \cdot [p_j]_S^\top - \mathsf{Id})$.

2 Solve the semidefinite program (SDP) (this step takes $n_{\mathsf{pub}}^{2\omega+1}$ via Jiang et al. (2020))

$$\max_{Z \succeq 0} \langle Z, \widetilde{\mathbf{M}} \rangle \text{ subject to } \mathrm{Tr}(Z) = 1, \sum_{i,j} |Z_{i,j}| \leq k_{\mathsf{pub}} \qquad (2)$$

Compute the top eigenvector $\widetilde{w}$ of $Z$.

3 **return** coordinates of the $k$ entries of $\widetilde{w}$ with the largest magnitudes.

---

*Then* $\mathbf{E}[\widetilde{\mathbf{M}}] = \frac{1}{2}[w]_S[w]_S^\top$.

*Proof.* First, it is obvious that the expectation of $\widetilde{\mathbf{M}}$ can be written as

$$\mathbf{E}[\widetilde{\mathbf{M}}] = \mathbf{E}_{p \sim \mathcal{N}(0, I_d)}[(\langle w, p \rangle^2 - 1) \cdot (p_S p_S^\top - \mathsf{Id})].$$

For any vector $v \in \mathbb{R}^n$ with $\|v\|_2 = 1$, we can compute $v^\top \mathbf{E}[\widetilde{\mathbf{M}}]v$

$$\begin{aligned}
v^\top \mathbf{E}[\widetilde{\mathbf{M}}]v &= v^\top \mathbf{E}_p[(\langle w, p \rangle^2 - 1) \cdot (p_S p_S^\top - \mathsf{Id})]v \\
&= \mathbf{E}_p[(\langle w, p \rangle^2 - 1) \cdot (\langle [v]_S, p \rangle^2 - 1)] \\
&= \mathbf{E}_p[(\langle w, p \rangle^2 - 1) \cdot (\|[v]_S\|_2^2 \langle [v]_S/\|[v]_S\|_2, p \rangle^2 - 1)] \\
&= \mathbf{E}_p[(\langle w, p \rangle^2 - 1) \cdot (\|[v]_S\|_2^2 \langle [v]_S/\|[v]_S\|_2, p \rangle^2 - \|[v]_S\|_2^2)] \\
&\quad + \mathbf{E}_p[(\langle w, p \rangle^2 - 1) \cdot (\|[v]_S\|_2^2 - 1)] \\
&=: A_1 + A_2
\end{aligned}$$

where the second step follows from $\|v\|_2^2 = 1$.

For the first term in the above equation, we have

$$\begin{aligned}
A_1 &= \mathbf{E}_p[(\langle w, p \rangle^2 - 1) \cdot (\|[v]_S\|_2^2 \langle [v]_S/\|[v]_S\|_2, p \rangle^2 - \|[v]_S\|_2^2)] \\
&= \|[v]_S\|_2^2 \mathbf{E}_p[(\langle w, p \rangle^2 - 1) \cdot (\langle [v]_S/\|[v]_S\|_2, p \rangle^2 - \|[v]_S\|_2^2)] \\
&= 2\|[v]_S\|_2^2 \mathbf{E}_p[\phi_2(\langle w, p \rangle) \cdot \phi_2(\langle [v]_S/\|[v]_S\|_2, p \rangle)] \\
&= 2\|[v]_S\|_2^2 \langle w, [v]_S/\|[v]_S\|_2 \rangle^2 \\
&= 2\langle w, [v]_S \rangle^2
\end{aligned}$$

where the third step follows from the fact that $w$ and $[v]_S/\|[v]_S\|_2$ are unit vectors, $\phi_2$ denotes the normalized degree-2 Hermite polynomial $\phi_2(z) \triangleq \frac{1}{\sqrt{2}}(z^2 - 1)$, and the last step follows from the standard fact that $\mathbf{E}_{g \sim \mathcal{N}(0, I_d)}[\phi_i(\langle g, v_1 \rangle)\phi_j(\langle g, v_2 \rangle)] = \langle v_1, v_2 \rangle^i$ if $i = j$ and $0$ otherwise.

For the second term, we have

$$A_2 = \mathbf{E}_p[(\langle w, p \rangle^2 - 1) \cdot (\|[v]_S\|_2^2 - 1)] = (\|[v]_S\|_2^2 - 1) \cdot \mathbf{E}_p[\langle w, p \rangle^2 - 1] = 0.$$

Thus, we have

$$A_1 + A_2 = 2\langle w, [v]_S \rangle^2.$$

In particular, for $v = [w]_S/\|[w]_S\|_2$, the above quantity is $2\|[w]_S\|_2^2$, while for $v \perp [w]_S$, the above quantity is 0. Thus we complete the proof. $\qquad \square$

Finally, we complete the proof of correctness of LEARNPUBLIC. Here we leverage the fact that we are running an SDP (the canonical SDP for sparse PCA) to show that as long as $d$ is at least polynomially large in $k_{\sf pub}$ and *logarithmically large in $n$*, with high probability we can recover $\mathrm{supp}([w]_S)$.

**Lemma B.4** (Learning the public coordinates). *For any $\delta > 0$, if $d \geq \mathrm{poly}(k_{\sf pub})/\log(n/\delta)$, then with probability at least $1 - \delta$ over the randomness of $\mathbf{X}$, we have that the coordinates output by* LEARNPUBLIC$(\{([p_j]_S, y_j)\}_{j\in[d]}$ *for* $y_j \triangleq |\langle p_j, w\rangle|$ *are exactly equal to* $\mathrm{supp}([w]_S)$.

*Proof.* Let $Z$ be the solution to the SDP in (2), and define $w_* \triangleq [w]_S/\|[w]_S\|$. Because $w_*$ is a feasible solution for the SDP, by optimality of $Z$ we get that

$$
\begin{aligned}
0 &\leq \langle Z - w_* w_*^\top, \widetilde{\mathbf{M}}\rangle \\
&= \langle Z - w_* w_*^\top, \mathbf{E}[\widetilde{\mathbf{M}}]\rangle + \langle Z - w_* w_*^\top, \widetilde{\mathbf{M}} - \mathbf{E}[\widetilde{\mathbf{M}}]\rangle \\
&= \frac{\|[w]_S\|^2}{2}\underbrace{\langle Z - w_* w_*^\top, w_* w_*^\top\rangle}_{\text{\textcircled{1}}} + \underbrace{\langle Z - w_* w_*^\top, \widetilde{\mathbf{M}} - \mathbf{E}[\widetilde{\mathbf{M}}]\rangle}_{\text{\textcircled{2}}},
\end{aligned}
\tag{3}
$$

where in the last step we used Lemma B.3.

Because $\|Z\|_F \leq \mathrm{Tr}(Z) = 1 = \|x_*\|$, we may upper bound ① by $-\frac{1}{2}\|Z - w_* w_*^\top\|_F^2$. For ②, note that because the entrywise $L_1$ norm of $Z$ and $x_* x_*^\top$ are both upper bounded by $k$, by Holder's we can upper bound ② by $2k_{\sf pub} \cdot \|\widetilde{\mathbf{M}} - \mathbf{E}[\widetilde{\mathbf{M}}]\|_{\max}$. Standard concentration (see e.g. Neykov et al. (2016)) implies that as long as $d \geq \log(n/\delta)/\eta^2$, then $\|\widetilde{\mathbf{M}} - \mathbf{E}[\widetilde{\mathbf{M}}]\|_{\max} \leq \eta$. We conclude from (3) that

$$
0 \leq -\frac{\|[w]_S\|^2}{4}\|Z - w_* w_*^\top\|_F^2 + 2k_{\sf pub}\eta,
$$

so $\|Z - w_* w_*^\top\|_F^2 \leq 8k_{\sf pub}\eta/\|[w]_S\|^2 \geq 8\eta k_{\sf pub}^2$, where in the last step we used that if $w$ has at least one public coordinate, then $\|[w]_S\|^2 \geq 1/k_{\sf pub}$. By Davis-Kahan, this implies that the top eigenvector $\widetilde{w}$ of $Z$ satisfies $\|\widetilde{w} - w_*\|^2 \leq 8\eta k_{\sf pub}^2$. As the nonzero entries of $w_*$ are at least $1/\sqrt{k_{\sf pub}}$, by taking $\eta = O(1/k_{\sf pub}^3)$ we ensure that $\|\widetilde{w} - w_*\|_\infty \leq \|\widetilde{w} - w_*\|_2 < 1/2\sqrt{k_{\sf pub}}$, so the largest entries of $\widetilde{w}$ in magnitude will be in the same coordinates as the nonzero entries of $w_*$. □

### B.2 RECOVERING THE GRAM MATRIX VIA FOLDED GAUSSIANS

We now turn to the second step of our overall recovery algorithm: recovering the $m \times m$ Gram matrix whose $(i, j)$-th entry is $\mathrm{supp}(w_i) \cap \mathrm{supp}(w_j)$. For this section and the next four sections, we will assume that $S = \emptyset$, i.e. that all images are private. For brevity, let $k \triangleq k_{\sf priv}$. This turns out to be without loss of generality. Given that in the case where $S \neq \emptyset$ we can recover the public coordinates of any selection vector using LEARNPUBLIC, passing to the case of general $S$ will be a simple matter of subtracting the contribution of the public coordinates from the entries of the Gram matrix obtained by GRAMEXTRACT to reduce to the case of $S = \emptyset$. We will elaborate on this in the final proof of Theorem B.1.

Given selection vectors $w_1, ..., w_m$, define the matrix $W \in \mathbb{R}^{m\times d}$ to have rows consisting of these vectors, so that the Gram matrix we are after is simply given by $WW^\top$. Recall that the $m \times d$ matrix whose rows consist of $y^{\mathbf{X},w_1}, ..., y^{\mathbf{X},w_m}$ can be written as

$$
\mathbf{Y} \triangleq \begin{pmatrix} |\langle p_1, w_1\rangle| & \cdots & |\langle p_d, w_1\rangle| \\ \vdots & \ddots & \vdots \\ |\langle p_1, w_m\rangle| & \cdots & |\langle p_d, w_m\rangle| \end{pmatrix},
$$

and as each entry of $\mathbf{X}$ is an independent standard Gaussian, the columns of $\mathbf{Y} \in \mathbb{R}_{\geq 0}^{m\times d}$ can be regarded as independent draws from $\mathcal{N}^{\sf fold}(0, WW^\top)$, where $W$ is defined above. Let $\mathbf{\Sigma}^{\sf fold}$ denote the covariance of this folded Gaussian distribution. It is known that one can recover information about the covariance $WW^\top$ of the original Gaussian distribution from the covariance $\mathbf{\Sigma}^{\sf fold}$ of its folded counterpart:

**Lemma B.5** (Page 7 in Kan & Robotti (2017))**.** *Given a Gaussian $\mathcal{N}(0, \Sigma)$, the covariance $\mathbf{\Sigma}^{\text{fold}} \in \mathbb{R}^{m \times m}$ of the corresponding folded Gaussian distribution $\mathcal{N}^{\text{fold}}(0, \Sigma)$ is given by $\mathbf{\Sigma}^{\text{fold}}_{i,i} = \Sigma_{i,i}$ and, for $i \neq j$,*

$$\mathbf{\Sigma}^{\text{fold}}_{i,j} = \Sigma_{i,j}(4\Phi_2(0, 0; \rho_{i,j}) - 1) + 4\Sigma^{1/2}_{i,i}\Sigma^{1/2}_{j,j}(1 - \rho^2_{i,j})\phi_2(0, 0; \rho_{i,j}) - \frac{2}{\pi}\Sigma^{1/2}_{i,i}\Sigma^{1/2}_{j,j}$$

*where $\rho_{i,j} \triangleq \Sigma_{i,j}/(\Sigma^{1/2}_{i,i}\Sigma^{1/2}_{j,j})$.*

We can apply Lemma B.5 in our specific setting to obtain the following relationship between $WW^{\top}$ and the covariance of $\mathcal{N}^{\text{fold}}(0, WW^{\top})$:

**Corollary B.6.** *If $\Sigma = WW^{\top} \in \mathbb{R}^{m \times m}$ for some matrix $W \in \mathbb{R}^{m \times n}$ where the rows of $W$ are unit vectors, then the covariance $\mathbf{\Sigma}^{\text{fold}} \in \mathbb{R}^{m \times m}$ of the corresponding folded Gaussian distribution $\mathcal{N}^{\text{fold}}(0, \Sigma)$ is given by*

$$\mathbf{\Sigma}^{\text{fold}}_{i,j} = \begin{cases} 1, & \text{if } i = j; \\ \Psi(\langle w_i, w_j \rangle), & \text{if } i \neq j. \end{cases}$$

*where $\Psi(z) \triangleq \frac{2}{\pi}(z \cdot \arcsin(z) + \sqrt{1 - z^2} - 1)$.*

*Proof.* Because the rows of $W$ are unit vectors, we have that $\Sigma_{i,j} = \rho_{i,j} = \langle w_i, w_j \rangle$ for all $i, j \in [m]$. To compute the off-diagonal entries of $\mathbf{\Sigma}^{\text{fold}}$, note that by definition of CDF and PDF,

$$\phi_2(0, 0; \langle w_i, w_j \rangle) = \frac{1}{2\pi\sqrt{1 - \langle w_i, w_j \rangle^2}}, \quad \Phi_2(0, 0; \langle w_i, w_j \rangle) = \frac{1}{4} + \frac{\arcsin\langle w_i, w_j \rangle}{2\pi}.$$

The claim follows. $\qquad\square$

---

**Algorithm 2:** GRAMEXTRACT($\{y^{\mathbf{X}, w_i}\}_{i \in [m]}, \eta$)

**Input:** InstaHide dataset $\{y^{\mathbf{X}, w_i}\}_{i \in [m]}$), accuracy parameter $\eta$
**Output:** Matrix $\mathbf{M}$ equal to the Gram matrix $k \cdot WW^{\top}$, scaled to have integer entries (see Lemma B.7)

1 $\eta^* \leftarrow O(\eta^2)$.
2 Let $z_1, ..., z_d \in \mathbb{R}^m$ be the vectors given by

$$(z_j)_i = y^{\mathbf{X}, w_i}_j.$$

for all $i \in [m], j \in [d]$.
3 Form the empirical estimates

$$\widehat{\mu} = \frac{1}{d}\sum_{i=1}^{d} z_i \qquad \widehat{\mathbf{\Sigma}} = \frac{1}{d}\sum_{i=1}^{d}(z_i - \widehat{\mu})(z_i - \widehat{\mu})^{\top}$$

and define $\widehat{\mathbf{\Sigma}}'$ to be the matrix obtained by applying the function $\text{clip}_{\eta^*}$ entrywise to $\widehat{\mathbf{\Sigma}}$.
4 Let $\widetilde{\mathbf{\Sigma}}$ be the matrix obtained by applying $\Psi^{-1}$ entrywise to $\widehat{\mathbf{\Sigma}}'$.
5 Let $\mathbf{\Sigma}^*$ denote the matrix obtained by entrywise rounding every entry of $\widetilde{\mathbf{\Sigma}}$ to the nearest multiple of $1/k$.
6 **return** $k \cdot \mathbf{\Sigma}^*$.

---

We now show that provided the number of pixels is moderately large, we can recover the matrix *exactly*, regardless of the choice of selection vectors $w_1, ..., w_m \in \mathbb{R}^n$. The full algorithm, GRAMEXTRACT, is given in Algorithm 2 above.

**Lemma B.7** (Extract Gram matrix)**.** *Suppose $d = \Omega(\log(m/\delta)/\eta^4)$. For random Gaussian image matrix $\mathbf{X}$ and arbitrary $w_1, ..., w_m \in \mathbb{S}^{d-1}_{\geq 0}$, let $\widetilde{\mathbf{\Sigma}}$ be the matrix computed in Step 4 of GRAMEXTRACT $(\{y^{\mathbf{X}, w_i}\}_{i \in [m]}, \eta)$, and let $\mathbf{\Sigma}^*$ be the output. Then with probability $1 - \delta$ over the randomness of $\mathbf{X}$, we have that $|\widetilde{\mathbf{\Sigma}}_{i,i'} - \langle w_i, w_{i'} \rangle| \leq \eta$ for all $i, i' \in [m]$. In particular, if $\eta = 1/2k$, the conditioned on this happening, $\mathbf{\Sigma}^* = k \cdot WW^{\top}$.*

To prove this, we will need the following helper lemma about $\Psi^{-1}$.

**Lemma B.8.** *There is an absolute constant $c > 0$ such that for any $0 < \eta < 1$ and $\widehat{z}, z \geq \eta$,*

$$|\Psi^{-1}(\widehat{z}) - \Psi^{-1}(z)| \leq \frac{c}{\sqrt{\eta}} \cdot |\widehat{z} - z|.$$

*Proof.* Noting that $\Psi'(z) = 2\arcsin(x)/\pi$, we get that the derivative of $\Psi^{-1}$ at $z$ is given by $\frac{1}{\Psi'(\Psi^{-1}(z))} = \frac{\pi}{2\arcsin(\Psi^{-1}(z))}$. One can verify numerically that for $0 \leq x \leq 1$, $\frac{x^2}{\pi} \leq \Psi(x) \leq \frac{1.2x^2}{\pi}$, so in particular $\sqrt{\pi z/1.2} \leq \Psi^{-1}(z) \leq \sqrt{\pi z}$. The derivative of $\Psi^{-1}$ at $z$ is therefore upper bounded by $O(1/\arcsin(\sqrt{\pi z/1.2})) \leq O(\sqrt{1.2/(\pi z)})$. In particular, for $z \geq \eta$, this is at most $O(1/\sqrt{\eta})$. In other words, over $\eta \leq z \leq 1$, $\Psi^{-1}$ is $O(1/\sqrt{\eta})$-Lipschitz as claimed. $\square$

Up to this point we have not used the randomness of the process generating the selection vectors $w_1, ..., w_m$. Note that without leveraging this, there exist choices of $W$ for which it is information-theoretically impossible to discern anything. Indeed, consider a situation where $w_1, ..., w_m \in \mathbb{S}_{\geq 0}^{d-1}$ have pairwise disjoint supports. In this case all we know is that the columns of $\mathbf{Y}$ are independent standard Gaussian vectors, as $WW^\top = \mathsf{Id}$. We now proceed to the most involved component of our proof, where we exploit the randomness of the selection vectors.

### B.3    SOLVING A LARGE SYSTEM OF EQUATIONS

In this section we show that if we can pinpoint a collection of selection vectors corresponding to all size-$k$ subsets of some set of $k + 2$ private images, then we can solve a certain system of equations to uniquely (up to sign) recover those private images. We will need the following basic notion corresponding to the fact that this system has only one unique solution, up to sign.

**Definition B.9** (Generic solution of system of equations). For any $m$ and any vector $v = (v_S)_{S \in \mathcal{C}_{[m]}^k}$ $\in \mathbb{R}^{\binom{m}{k}}$, we say that $v$ is *generic* if there are at most two solutions to the system

$$\left|\sum_{i \in S} a_i\right| = v_S \qquad \forall S \in \mathcal{C}_{[m]}^k$$

in the variables $\{a_i\}_{i \in [m]}$. Note that there are exactly two solutions $\{a_i'\}$ and $\{a_i''\}$ to this system if and only if $a_i' = -a_i''$ for all $i \in [m]$ and $a_i' \neq 0$ for some $i \in [m]$.

We now show that for Gaussian images, the abovementioned system of equations almost surely has a unique solution up to sign.

**Lemma B.10** (Vector of Gaussian subset sums is generic). *Let $g_1, ..., g_m$ be independent draws from $\mathcal{N}(0, 1)$. For any $m$ satisfying $m \geq k + 2$, the vector $v = (v_S)_{S \in \mathcal{C}_{[m]}^k}$ given by $v_S \triangleq \sum_{i \in S} g_i$ is generic almost surely (with respect to the randomness of $g_1, ..., g_m$).*

*Proof.* First note that the entries of $v$ are all nonzero almost surely. For $v$ to not be generic, there must exist another vector $v'$ whose entrywise absolute value satisfies $|v| = |v'|$ but for which $v' \neq v, -v$ and for which there exists $h_1, ..., h_m$ satisfying $\sum_{i \in S} h_i = v_S'$ for all $S \in \mathcal{C}_{[m]}^k$. This would imply there exist indices $S, T$ for which $v_S' = v_S$ and $v_T' = -v_T$.

By the assumption that $m \geq k + 2$ (and recalling that $k > 1$ in our setup), we have that $\binom{m}{k} > m$. In particular, the set of vectors $w = (w_S)_{S \in \mathcal{C}_{[m]}^k}$ for which there exist numbers $\{g_i'\}$ such that $w_S = \sum_{i \in S} g_i'$ for all $S$ is a proper subspace $U$ of $\mathbb{R}^{\binom{m}{k}}$. Let $\ell_1, ..., \ell_a$ be a basis for the set of vectors $\ell$ satisfying $\langle \ell, w \rangle = 0$ for all $w \in U$. Note that there is at least one nonzero generic vector in $U$, for instance, the vector $w^*$ given by $w_S^* = \mathbb{1}[i \in S]$ (here we again use the fact that $m \geq k+2$).

Letting $\mathbf{D} \in \mathbb{R}^{\binom{m}{k} \times \binom{m}{k}}$ denote the diagonal matrix whose $S$-th diagonal entry is equal to $v_S/v_S'$, note that the existence of $h_1, ..., h_m$ above implies that $v$ additionally satisfies $\langle \mathbf{D}\ell_i, v \rangle = 0$ for all $i \in [a]$. But there must be some $i$ for which $\mathbf{D}\ell_i$ does not lie in the span of $\ell_1, ..., \ell_a$, or else we would conclude that for any $w \in U$, the vector $w'$ whose $S$-th entry is $w_S \cdot v_S/v_S'$ would also lie

in $U$. Because of the existence of indices $S, T$ for which $v'_S = v_S$ and $v'_T = -v_T$, we know that $w \neq w', -w'$, so we would erroneously conclude that $w$ is not generic for any $w \in U$, contradicting the fact that the vector $w^*$ defined above is generic.

We conclude that there is some $i$ for which $\mathbf{D}\ell_i$ lies outside the span of $\ell_1, \ldots, \ell_a$. But then the fact that $\langle \mathbf{D}\ell_i, v \rangle = 0$ for this particular $i$ implies that the variables $g_i$ satisfy some nontrivial linear relation. This almost surely cannot be the case because $g_1, \ldots, g_m$ are independent draws from $\mathcal{N}(0,1)$. □

### B.4 Locating a Set of Useful Selection Vectors

In the previous section we showed that we just need to find a set of selection vectors from among the rows of $W$ that correspond to size-$k$ subsets of some set of $k + 2$ private images. Here we show that such a collection of selection vectors is uniquely identified, up to trivial ambiguities, by their pairwise inner products.

**Lemma B.11** (Uniquely identifying a family of subsets). *Let $\mathcal{F} = \{T_S\}_{S \in \mathcal{C}^k_{[k+2]}}$ be a collection of subsets of $[n]$ for which $|T_S \cap T_{S'}| = |S \cap S'|$ for all $S, S' \in \mathcal{C}^k_{[k+2]}$. Then there is some subset $U \subseteq [n]$ of size $k + 2$ for which $\{T_S\} = \mathcal{C}^k_U$ as (unordered) sets.*

| 1 | 2 | 3 | 4 | | |
|---|---|---|---|---|---|
| | | 3 | 4 | 5 | 6 |
| 1 | | 3 | 4 | 5 | |
| 1 | | 3 | 4 | | 6 |
| | 2 | 3 | 4 | 5 | |
| | 2 | 3 | 4 | | 6 |
| 1 | 2 | 3 | | 5 | |
| 1 | 2 | | 4 | 5 | |
| 1 | 2 | 3 | | | 6 |
| 1 | 2 | | 4 | | 6 |
| 1 | | 3 | | 5 | 6 |
| 1 | | | 4 | 5 | 6 |
| | 2 | 3 | | 5 | 6 |
| | 2 | | 4 | 5 | 6 |
| 1 | 2 | | | 5 | 6 |

Table 1: Illustration of the sequence of subsets constructed in the proof of Lemma B.11 for $k = 4$. Red and blue denote $S_0$ and $S_1$, purple denotes $S_{a,b}$ for $a \in \{1, 2\}$, $b \in \{k + 1, k + 2\}$, green denotes the $4k - 8$ sets $S''$, and gold denotes the $\binom{k-2}{k-4} = 1$ set $S'''$.

*Proof.* For the reader's convenience, we illustrate the sequence of subsets constructed in the following proof in Table 1.

Suppose without loss of generality that $\mathcal{F}$ contains the sets $S_{1,2} \triangleq \{1, \ldots, k\}$ and $S_{k+1,k+2} \triangleq \{3, \ldots, k + 2\}$ (the indexing will become clear momentarily). We will show that $\{T_S\} = \mathcal{C}^k_U$ for $U = [k+2]$.

Let $S^* \triangleq S_0 \cap S_1$. For any $S' \in \mathcal{C}^k_{[k+2]}$ satisfying $|S_0 \cap S'| = |S_1 \cap S'| = k - 1$, observe that $S'$ must contain $S^*$ and one element from each of $S_0 \backslash S_1 = \{1, 2\}$ and $S_1 \backslash S_0 = \{k + 1, k + 2\}$, so there are four such choices of $S'$, call them $\{S_{a,b}\}_{a \in \{1,2\}, b \in \{k+1,k+2\}}$, and $\mathcal{F}$ must contain all of them.

Now consider any subset $S'' \subset [k + 2]$ for which, for some $b \neq b' \in \{k + 1, k + 2\}$, we have that $|S'' \cap S_{1,2}| = |S'' \cap S_{1,b}| = |S'' \cap S_{2,b}| = k - 1$, and $|S'' \cap S_{k+1,k+2}| = |S'' \cap S'_{1,b'}| = |S'' \cap S'_{2,b'}| = k - 2$. Observe that it must be that $|S'' \cap S^*| = k - 3$ and that $S''$ contains $\{1, 2\}$, so there are $2 \cdot \binom{k-2}{k-3} = 2k - 4$ such choices of $S''$, and $\mathcal{F}$ must contain all of them. We can similarly consider $S''$ for which, for some $a \neq a' \in \{1, 2\}$, we have that $|S'' \cap S_{k+1,k+2}| = |S'' \cap S_{a,k+1}| = $

$|S'' \cap S_{a,k+2}| = k - 1$, and $|S'' \cap S_{1,2}| = |S'' \cap S'_{a',k+1}| = |S'' \cap S'_{a',k+2}| = 2k - 4$, for which there are again $2k - 4$ choices of $S''$, and $\mathcal{F}$ must contain all of them.

Alternatively, if $\mathcal{F}$ contained $k - 2$ subsets $S''$ satisfying $|S'' \cap S_{1,2}| = |S'' \cap S_{b,k+1}| = |S'' \cap S_{b,k+2}| = k - 1$ for some $b \in \{1, 2\}$, then it would have to be that any such $S''$ contains the $k - 1$ elements of $\{b, 3, \ldots, k\}$, and therefore the intersection between any pair of such $S''$ must be equal to $k - 1$, violating the constraint that $|T_S \cap T_{S'}| = |S \cap S'|$ for all $S, S' \in \mathcal{C}^k_{[k+2]}$. The same reasoning applies to rule out the case where $\mathcal{F}$ contains $k - 2$ subsets $S''$ satisfying $|S'' \cap S_{k+1,k+2}| = |S'' \cap S_{1,b}| = |S'' \cap S_{2,b}| = k - 1$ for some $b \in \{k + 1, k + 2\}$.

Finally, consider the set of all subsets $S'''$ distinct from the ones exhibited thus far, and for which $|S''' \cap S_0| = |S''' \cap S_1| = |S''' \cap S_{a,b}| = k - 2$ for all $a \in \{1, 2\}, b \in \{k + 1, k + 2\}$ and $|S''' \cap S''|$ for at least one of the $4k - 8$ subsets constructed two paragraphs above. Observe that any $S'''$ distinct from the ones exhibited thus far which satisfies the first constraint must either contain $S^*$ and two elements outside of $\{1, \ldots, k + 4\}$, or must satisfy $|S''' \cap S^*| = k - 4$ and contain $\{1, 2, k + 1, k + 2\}$. In the former case, such an $S'''$ would violate the second constraint. As for the latter case, there are $\binom{k-2}{k-4}$ such choices of $S'''$, and $\mathcal{F}$ must therefore contain all of them. We have now produced $4k - 2 + \binom{k-2}{k-4} = \binom{k+2}{k}$ unique subsets, all belonging to $\mathcal{C}^k_{[k+2]}$, and $\mathcal{F}$ is of size $\binom{k+2}{k}$, concluding the proof. □

## B.5 EXISTENCE OF A FLORAL SUBMATRIX

Recall the notion of a *floral* submatrix from Definition 3.1. In this section we show that with high probability $\mathbf{M}$ contains a floral principal submatrix. In the language of sets, this means that with high probability over a sufficiently long sequence of randomly chosen size-$k$ subsets of $[n]$, there is a collection of $\binom{k+2}{k}$ subsets in the sequence which together comprise all size-$k$ subsets of some $U \subseteq [n]$ of size $k + 2$. Quantitatively, we have the following:

**Lemma B.12** (Existence of a floral submatrix). *Let $m \geq \Omega(k^{O(k^3)} n^{k - \frac{2}{k+1}})$. If sets $T_1, \ldots, T_m$ are independent draws from the uniform distribution over $\mathcal{C}^k_n$, then with probability at least $9/10$, there is some $U \in \mathcal{C}^{k+2}_{[n]}$ for which every element of $\mathcal{C}^k_U$ is present among $T_1, \ldots, T_m$.*

*Proof.* Let $L = \binom{k+2}{k} = \frac{1}{2}(k + 2)(k + 1)$. Define

$$Z \triangleq \sum_{i_1 < \cdots < i_L \in [m]} \mathbb{1} \left[ \{T_{i_1}, \ldots, T_{i_L}\} = \mathcal{C}^k_U \text{ for some } U \in \mathcal{C}^{k+2}_{[n]} \right].$$

By linearity of expectation, $\mathbf{E}[Z]$ is equal to $\binom{m}{L}$ times the probability that $\{T_1, \ldots, T_L\} = \mathcal{C}^k_U$ for some $U \in \mathcal{C}^{k+2}_{[n]}$. The latter probability is equal to $\binom{n}{k+2} \cdot L! \cdot \binom{n}{k}^{-L}$, so we conclude that

$$\begin{aligned}
\mathbf{E}[Z] &= \binom{m}{L} \cdot \binom{n}{k+2} \cdot L! \cdot \binom{n}{k}^{-L} \\
&\geq m^L \cdot \frac{n^{k+2}}{n^{kL}} \cdot \frac{L! \cdot (k!)^L}{L^L \cdot (k+2)^{k+2}} \\
&\geq \Omega\left(m^L n^{k+2-kL}\right) \geq \Omega(1),
\end{aligned}$$

where in the penultimate step we used that $\frac{L! \cdot (k!)^L}{L^L \cdot (k+2)^{k+2}}$ is nonnegative and increasing over $k \geq 2$, and in the last step we used that $m \geq \Omega\left(n^{k - \frac{2}{k+1}}\right)$.

We now upper bound $\mathbf{E}[Z^2]$. Consider a pair of distinct summands $(i_1, \ldots, i_L)$ and $(i'_1, \ldots, i'_L)$. Without loss of generality, we may assume these are $(1, \ldots, L)$ and $(s+1, \ldots, L)$ for some $0 \leq s \leq L$. In order for $\{T_1, \ldots, T_L\} = \mathcal{C}^k_U$ and $\{T_{L-s+1}, \ldots, T_{2L-s+1}\} = \mathcal{C}^k_{U'}$ for some $U, U' \in \mathcal{C}^{k+2}_{[n]}$, it must be that $\{T_{L-s+1}, \ldots, T_L\} = \mathcal{C}^k_{U \cap U'}$. Note that if $|U \cap U'| = k + 2$, then $U = U'$ and therefore $s$ must be 0. So if $s > 0$, it must be that $|U \cap U'| \in \{k, k + 1\}$.

In either case, the probability that $\{T_1, ..., T_{L-s+1}\} = \mathcal{C}_U^k \backslash \mathcal{C}_{U \cap U'}^k$, $\{T_{L+1}, ..., T_{2L-s+1}\} = \mathcal{C}_U^k \backslash \mathcal{C}_{U \cap U'}^k$, and $\{T_{L-s+1}, ..., T_L\} = \mathcal{C}_{U \cap U'}^k$ is

$$(L-s)!^2 \cdot s! \cdot \binom{n}{k}^{-2L+s} \leq L!^2 \cdot (k/n)^{2kL-ks}$$

If $|U \cap U'| = k$, then $s$ must be 1, and there are

$$\binom{n}{k} \cdot \binom{n-k-2}{2} \cdot \binom{n-k-4}{2} \leq n^{k+4}$$

choices for $(U, U')$. If $|U \cap U'| = k+1$ then $s$ must be $k+1$ and there are and there are

$$\binom{n}{k+1} \cdot (n-k-1) \cdot (n-k-2) \leq n^{k+3}$$

choices for $(U, U')$.

Finally, note that there are $\binom{m}{L}$ pairs of summands $(i_1, ..., i_l), (i'_1, ..., i'_L)$ for which $s = 0$ (namely the ones for which $i_j = i'_j$ for all $j$), $m \cdot \binom{m-1}{L-1} \cdot \binom{m-L}{L-1} \leq \Theta(m)^{2L-1} \cdot L!^2$ pairs for which $s = 1$, and $\binom{m}{k+1} \cdot \binom{m-k-1}{L-k-1} \cdot \binom{m-L}{L-k-1} \leq \Theta(m)^{2L-k-1} \cdot L!^2$ for which $s = k+1$. Putting everything together, we conclude that

$$\mathbf{E}[Z^2] = \mathbf{E}[Z] + \Theta(m)^{2L-1} \cdot L!^4 \cdot n^{k+4} \cdot (k/n)^{2kL-k} + \Theta(m)^{2L-k-1} \cdot L!^4 \cdot n^{k+3} \cdot (k/n)^{2kL-k(k+1)}$$

$$\leq \mathbf{E}[Z]^2 \cdot \left(1 + O(1/m) \cdot L!^4 \cdot k^{2kL-k} + O(1/m^{k+1}) \cdot L!^4 \cdot k^{2kL-k(k+1)} \cdot n^{k^2-1}\right)$$

$$\leq (1.01 \, \mathbf{E}[Z])^2,$$

where in the last step we used that $L \leq k^2$ and that $n^{k^2-1}/m^{k+1} \leq 1$ because $m \geq k^{\Omega(k^3)} n^{k-1}$.

By Paley-Zygmund, we conclude that

$$\mathbb{P}[Z > 0.01 \, \mathbf{E}[Z]] \geq 0.99^2 \cdot \frac{\mathbf{E}[Z]^2}{\mathbf{E}[Z^2]} \geq 9/10,$$

as desired, upon picking constant factors appropriately. $\qquad\square$

Lemma B.12 implies that with probability at least 9/10 over the randomness of the mixup vectors $w_1, ..., w_m$, if $m \geq \Omega(k^{O(k^3)} n^{k-\frac{2}{k+1}})$, then there is a subset of $[m]$ for which the corresponding principal submatrix of $WW^\top$ is floral. By Lemma B.7, with high probability $\mathbf{M} = k \cdot WW^\top$, so this is also the case for the output of GRAMEXTRACT.

### B.6 FINDING A FLORAL SUBMATRIX

As mentioned in Section 3, to find a floral principal submatrix of $\mathbf{M}$, one option is to enumerate over all subsets of size $\binom{k+2}{k}$ of $[m]$, which would take $n^{O(k^3)}$ time. We now give a much more efficient procedure for identifying a floral principal submatrix of $\mathbf{M}$, whose runtime is dominated by the time it takes to write down the entries of $\mathbf{M}$. At a high level, the reason we can obtain such dramatic savings is that the underlying graph defined by the large entries of $WW^\top$ is quite sparse, i.e. vertices of the graph typically have degree independent of $k$.

We will need the following basic notion:

**Definition B.13.** Given $i \in [m]$ and integer $0 \leq t \leq k$, let $\mathcal{N}_i^t \triangleq \{j : \langle w_i, w_j \rangle = t/k\}$. For any $j \in \mathcal{N}_i^t$, we refer to $i$ and $j$ as $t$-neighbors (this relation is obviously commutative).

We will also need the following helper lemmas establishing certain deterministic regularity conditions that $WW^\top$ will satisfy with high probability.

**Lemma B.14** (Hypergraph sparsity). *For any $\delta > 0$, if $m \geq n^{k-1} \log(1/\delta)$, then with probability at least $1 - 2m\delta$ over the randomness of $w_1, ..., w_m$, we have that for every $j \in [m]$, there are at most $O(m \cdot k^{k+1} \cdot n^{1-k})$ $(k-1)$-neighbors of $j$, and at most $O(m \cdot k^{k+2} \cdot n^{2-k})$ $(k-2)$-neighbors of $j$.*

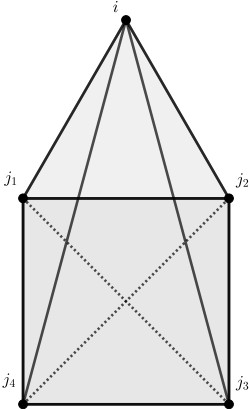

Figure 2: Illustration of a *house* $(i; j_1, j_2, j_3, j_4)$ where the solid lines indicate an entry of $k - 1$ in $\mathbf{M}$, while the dotted lines indicate an entry of $k - 2$.

*Proof.* We will union bound over $j \in [m]$, so without loss of generality fix $j = 1$ in the argument below. Let $X_{j'}$ (resp. $Y_{j'}$) denote the indicator for the event that 1 and $j'$ are $(k-1)$-neighbors (resp. $(k-2)$-neighbors). As $w_{j'}$ is sampled independently of $w_1$, conditioned on $w_1$ we know that $X_{j'}$ is a Bernoulli random variable with expectation $\mathbf{E}[X_{j'}] = \frac{k(n-k)}{\binom{n}{k}} \le n^{1-k} \cdot k^{k+1}$, where the factor of $k(n-k)$ comes from the number of ways to pick $\mathrm{supp}(w_1) \backslash \mathrm{supp}(w_{j'})$ and $\mathrm{supp}(w_{j'}) \backslash \mathrm{supp}(w_1)$. Similarly, $Y_{j'}$ is a Bernoulli random variable with expectation $\mathbf{E}[Y_{j'}] = \frac{\binom{k}{2}\binom{n-k}{2}}{\binom{n}{k}} \le n^{2-k} \cdot k^{k+2}$. By Chernoff, we conclude that $\sum_{j'>2} X_{j'} > 2n^{1-k} \cdot k^{k+1}$ with probability at most

$$\exp\left(-m \cdot D(\mathrm{Ber}(2n^{1-k} \cdot k^{k+1}) \| \mathrm{Ber}(n^{1-k} \cdot k^{k+1}))\right) \le \exp(-\Omega(mn^{1-k} \cdot k^{k+1}))$$
$$\le \exp(-\Omega(mn^{1-k})),$$

from which the first claim follows. Similarly by Chernoff, $\sum_{j'>2} Y_{j'} > 2n^{2-k} \cdot k^{k+2}$ with probability at most

$$\exp\left(-m \cdot D(\mathrm{Ber}(2n^{2-k} \cdot k^{k+2}) \| \mathrm{Ber}(n^{2-k} \cdot k^{k+2}))\right) \le \exp(-\Omega(mn^{2-k} \cdot k^{k+2}))$$
$$\le \exp(-\Omega(mn^{2-k})),$$

from which the second claim follows.

$\square$

**Definition B.15.** Given symmetric matrix $\mathbf{M} \in \mathbb{Z}^{m \times m}$ and distinct indices $i, j_1, ..., j_4 \in [m]$ for which $j_1 < j_4$, we say that $(i; j_1, \ldots, j_4)$ is a *house* (see Figure 2) if for all $1 \le a < b \le 4$, $\mathbf{M}_{j_a,j_b} = k - 1$ if $(a,b) \in \{(1,2),(2,1),(2,3),(3,4),(1,4)\}$ and $\mathbf{M}_{j_a,j_b} = k - 2$ otherwise, and furthermore $\mathbf{M}_{i,j_a} = k - 1$ for all $a \in [4]$.

**Lemma B.16** (Upper bounding the number of houses). *If $m \ge \Omega(n^{2k/3})$, then with probability at least $9/10$ over the randomness of $w_1, \ldots, w_m$, there are at most $O(k^{5k} \cdot m^5 \cdot n^{-4k+2})$ houses in $\mathbf{M}$.*

*Proof.* Define

$$Z \triangleq \sum_{i,j_1,\ldots j_4 \text{ distinct}, j_1 < j_4} \mathbb{1}\left[(i; j_1, \ldots, j_4) \text{ is a house}\right].$$

By linearity of expectation, $\mathbf{E}[Z]$ is equal to $m \cdot \binom{m-1}{4} \le m^5$ times the probability that $(1; 2, 3, 4, 5)$ is a house. Note that the only way for $(1; 2, 3, 4, 5)$ to be a house is if there are disjoint subsets $S_1, S - 2, T \subseteq [n]$ of size 2, 2, and $k - 2$ respectively such that $w_1$ is supported on $S \cup T$ and each of $w_2, \ldots, w_5$ is supported on $\{s_1, s_2\} \cup T$ where $s_1 \in S_1$, $s_2 \in S_2$. There are

$O\left(\binom{n}{k-2} \cdot \binom{n}{2}^2\right) \le n^{k+2}$ such choices of $(S_1, S_2, T)$, and for each is an $O(\binom{n}{k}^{-5})$ chance that the supports of $w_1, \ldots, w_5$ correspond to a given $(S_1, S_2, T)$, so we conclude that

$$\mathbf{E}[Z] = O\left(m^5 \cdot n^{k+2} \cdot \binom{n}{k}^{-5}\right) \le O(k^{5k} \cdot m^5 \cdot n^{-4k+2}).$$

We now upper bound $\mathbf{E}[Z^2]$. Consider a pair of distinct summands $(i; j_1, \ldots, j_4)$ and $(i'; j_1', \ldots, j_4')$. Recall that they correspond to some $(S_1, S_2, T)$ and $(S_1', S_2', T')$ respectively. Note that if these tuples overlap in any index (e.g. $(1; 2, 3, 4, 5)$ and $(6; 1, 7, 8, 9)$), then $|(S_1 \cup S_2 \cup T) \cap (S_1' \cup S_2' \cup T')| \ge k$. There are at most

$$O\left(\binom{n}{k} \cdot \binom{n-k}{2} \cdot \binom{n-k-2}{2} + \binom{n}{k+1} \cdot \binom{n-k}{1} \cdot \binom{n-k-1}{1} + \binom{n}{k+2}\right) \le O(n^{k+4})$$

pairs of sets $U, U' \subseteq [n]$ of size $k+2$ with intersection of size at least $k$, and given a set $U$ of size $k+2$, there are $O\left(\binom{k+2}{k-2}\right) \le \text{poly}(k)$ ways of partitioning $U$ into three disjoint sets of size 2, 2, and $k-2$ respectively. We conclude that any pair of distinct summands in the expansion of $\mathbf{E}[Z^2]$ altogether contributes at most $\text{poly}(k) \cdot O(n^{k+4}) \cdot \binom{n}{k}^{-b} \le k^{10k} \cdot n^{-(b-1)k+4}$, where $6 \le b \le 10$ is the number of distinct indices within the tuples $(i; j_1, \ldots, j_4)$ and $(i'; j_1', \ldots, j_4')$. For any $b$, there are $\binom{m}{5} \cdot \binom{m-5}{b-5} \le m^b$ such pairs of tuples.

In the special case where $b = 6$, we will use a slightly sharper bound by noting that then, it must be that $S_1 \cup S_2 \cup T$ and $S_1' \cup S_2' \cup T'$ are identical, in which case we can improve the above bound of $O(n^{k+4})$ for the number of pairs $U, U'$ to $O(n^{k+2})$.

We conclude that

$$\mathbf{E}[Z^2] \le \mathbf{E}[Z] + k^{10k} m^6 \cdot n^{-5k+2} + \sum_{b=7}^{10} m^b \cdot n^{-(b-1)k+4} \le O(k^{10k} \cdot m^{10} \cdot n^{-8k+4}).$$

where in the last step we used the fact that $m \ge O(n^{2k/3})$ and $k \ge 2$ to bound the summands corresponding to $b = 6$ and $b = 7$. Finally, by our bounds on $\mathbf{E}[Z]$ and $\mathbf{E}[Z^2]$, we conclude by Chebyshev's that with probability at least $9/10$, there are most $2\mathbf{E}[Z] \le O(k^{5k} \cdot m^5 \cdot n^{-4k+2})$ houses in $\mathbf{M}$. $\qquad\square$

**Lemma B.17** (Finding a floral submatrix). *Suppose $m = \Omega(n^{k-\frac{2}{k+1}})$. With probability at least $3/4$, FINDFLORALSUBMATRIX($\mathbf{M}$) runs in time $O(n^{2k-\frac{4}{k+1}} \cdot \exp(\text{poly}(k)))$ and outputs $\binom{k+2}{k} \times \binom{k+2}{k}$-sized subset $\mathcal{I} \subseteq [m]$ indexing a principal submatrix of $\mathbf{M}$ which is floral, together with a function $F : \mathcal{I} \to \mathcal{C}_{[k+2]}^k$ such that $\mathbf{M}_{j,j'} = |F(j) \cap F(j')|$ for all $j, j' \in \mathcal{I}$.*

*Proof.* The proof of correctness essentially follows immediately from the proof of Lemma B.11, while the runtime analysis will depend crucially on the sparsity of the underlying weighted graph defined by $\mathbf{M}$, as guaranteed by Lemmas B.14 and B.16. Henceforth, condition on the events of those lemmas holding, which will happen with probability at least $3/4$.

First note that if one reaches as far as Step 20 in FINDFLORALSUBMATRIX, then by the proof of Lemma B.11, the $\mathcal{I}$ produced in Step 22 indexes a principal submatrix of $\mathbf{M}$ which is floral. The recursive call in Step 24 is applied to a submatrix of $\mathbf{M}$ whose size is independent of $n$, and it is evident that the time expended past that point is no worse than some $\exp(\text{poly}(k))$, and inductively we know that the resulting $F$ produced in Step 25 when the recursion is complete correctly maps indices $j \in [m]$ to subsets in $\mathcal{C}_{[k+2]}^k$ such that $\mathbf{M}_{j,j'} = |F(j) \cap F(j')|$ for all $j, j' \in \mathcal{I}$.

To carry out the rest of the runtime analysis, it suffices to bound the time expended leading up to the recursive call. Consider any house $(i_0; j_1, j_2, j_3, j_4)$ encountered in Step 5. First note that one can compute $\bigcap_{a=1}^4 \mathcal{N}_{j_a}^{k-1}$ with a basic hash table, so because the first part of Lemma B.14 tells us that with high probability, $|\mathcal{N}_{j_a}^{k-1}| \le O(m \cdot k^{k+1} \cdot n^{1-k})$ for all $a \in [4]$, Step 5 only requires $O(m \cdot k^{k+1} \cdot n^{1-k})$ time. Similarly, for each of the $O(1)$ possibilities in the loop in Step 14, it takes $O(m \cdot k^{k+1} \cdot n^{1-k})$ time to enumerate over $(k-1)$-neighbors of $i_z, i_\alpha, i_\beta$ in Step 15 and, by

the second part of Lemma B.14, $O(m \cdot k^{k+2} \cdot n^{2-k})$ time to enumerate over $(k-2)$-neighbors of $i_{1-z}, i_\gamma, i_\delta$, and it takes $\text{poly}(k)$ to check that the resulting indices $i''$ are not all $(k-1)$-neighbors of each other. And once more, in Step 20 it takes $O(m \cdot k^{k+2} \cdot n^{2-k})$ time to enumerate over all indices which are $(k-2)$ neighbors of $i_0, i_1$ and of every $i'' \in \mathcal{I}''$.

We conclude that for every house $(i_0; j_1, j_2, j_3, j_4)$, FINDFLORALSUBMATRIX expends at most $O(m \cdot k^{k+2} \cdot n^{2-k})$ time checking whether the house can be expanded into a set of indices corresponding to a floral principal submatrix of $\mathbf{M}$. Note that for any $(i_0; j_1, j_2, j_3, j_4)$ encountered in Step 4 which is *not* a house, the algorithm expends $O(1)$ time. As $|\mathcal{N}_{i_0}^{k-1}| \leq O(m \cdot n^{1-k} \cdot k^{k+1})$ with high probability for any $i_0$, there are most $O(m \cdot m^4 \cdot n^{4-4k} \cdot k^{4k+4}) \leq O(m^5 \cdot n^{4-4k} \cdot k^{4k+4})$ such tuples which are not houses.

And because Lemma B.16 tells us that with high probability there are $O(k^{5k} \cdot m^5 \cdot n^{-4k+2})$ houses in $\mathbf{M}$, FINDFLORALSUBMATRIX outputs None with low probability. In particular, given that any single house $(i_0; j_1, j_2, j_3, j_4)$ expends $O(m \cdot k^{k+2} \cdot n^{2-k})$ time from Step 9 all the way potentially to Step 24, we conclude that the houses contribute a total of at most $O(k^{5k} \cdot m^5 \cdot n^{-4k+2} \cdot m \cdot k^{k+2} \cdot n^{2-k}) \leq O(m^6 \cdot n^{4-5k} \cdot k^{6k+2})$ to the runtime.

Putting everything together, we conclude that FINDFLORALSUBMATRIX runs in time

$$O\left(m^5 \cdot n^{4-4k} \cdot k^{4k+4} + m^6 \cdot n^{4-5k} \cdot k^{6k+2}\right) = O\left(n^{k+4-\frac{10}{k+1}} \cdot k^{O(k)}\right).$$

Lastly, note that $k + 4 - \frac{10}{k+1} \leq 2k - \frac{4}{k+1}$ whenever $k \geq 2$, completing the proof. $\qquad\square$

## B.7 Putting Everything Together

We are now ready to conclude the proof of correctness of our main algorithm, LEARNPRIVATEIMAGE.

*Proof.* By Lemma B.4, the subsets $S_i$ computed in Step 3 correctly index the public coordinates of $w_i$. By Lemma B.7, with high probability over the randomness of $\mathbf{X}$, the matrix $\mathbf{M}$ formed from GRAMEXTRACT in Step 1 of LEARNPRIVATEIMAGE is exactly equal to the Gram matrix $WW^\top$, so after Step 5 and Step 6, $\mathbf{M}$ is equal to the Gram matrix of the vectors $[w_1]_{S^c}, \ldots, [w_m]_{S^c}$, i.e. the restrictions of the selection vectors to the private coordinates. We are now in a position to apply the results of Sections B.3, B.4, B.5, and B.6.

By Lemma B.17, with high probability the output $\mathcal{I}, F$ of FINDFLORALSUBMATRIX in Step 7 satisfies that 1) the principal submatrix of $\mathbf{M}$ indexed by $\mathcal{I}$, a set of indices of size $\binom{k_{\text{priv}}+2}{k_{\text{priv}}}$, is floral, and 2) the function $F : \mathcal{I} \rightarrow \mathcal{C}_{[k_{\text{priv}}+2]}^{k_{\text{priv}}}$ satisfies that $|F(i) \cap F(j)| = \mathbf{M}_{i,j}$ for all $i, j \in \mathcal{I}$. By Lemma B.11, because the principal submatrix indexed by $\mathcal{I}$ is floral, there exists some subset $U \subseteq [n]$ of size $k_{\text{priv}} + 2$ for which the supports of the mixup vectors $w_j$ for $j \in \mathcal{I}$ are all the subsets of $U$ of size $k_{\text{priv}}$. Finally, by Lemma B.10 and the fact that the entries of $\mathbf{X}$ are independent Gaussians, for every pixel index $\ell \in [d]$, the solution $\{\widetilde{x}_i^{(\ell)}\}$ to the system in Step 8 satisfies that there is some column $x$ of the original private image matrix $\mathbf{X}$ such that for every $i \in [k_{\text{priv}} + 2]$, $\widetilde{x}_i^{(\ell)}$ is, up to signs, equal to the $\ell$-th pixel of $x$.

Note that the runtime of LEARNPRIVATEIMAGE is dominated by the operations of forming the matrix $\mathbf{M}$ and running FINDFLORALSUBMATRIX, which take time $O(m^2)$ by Lemma B.17. $\qquad\square$

## B.8 Example of a Floral Submatrix

**Example B.18.** For $k = 2$, the following $6 \times 6$ matrix, after dividing every entry by $k$, is floral:

|        | $\{1,3\}$ | $\{2,4\}$ | $\{1,4\}$ | $\{1,2\}$ | $\{3,4\}$ | $\{2,3\}$ |
|--------|-----------|-----------|-----------|-----------|-----------|-----------|
| $\{1,3\}$ | 2 | 0 | 1 | 1 | 1 | 1 |
| $\{2,4\}$ | 0 | 2 | 1 | 1 | 1 | 1 |
| $\{1,4\}$ | 1 | 1 | 2 | 1 | 1 | 0 |
| $\{1,2\}$ | 1 | 1 | 1 | 2 | 0 | 1 |
| $\{3,4\}$ | 1 | 1 | 1 | 0 | 2 | 1 |
| $\{2,3\}$ | 1 | 1 | 0 | 1 | 1 | 2 |

---

**Algorithm 3:** FINDFLORALSUBMATRIX($\mathbf{M}, k, r$)

---

**Input:** Query access to matrix $\mathbf{M} \in \mathbb{R}^{M \times M}$, sparsity level $k$

**Output:** $\binom{k+2}{k} \times \binom{k+2}{k}$-sized subset $\mathcal{I} \subseteq [M]$, function $F : \mathcal{I} \to \mathcal{C}^k_{[k+2]}$ (Lemma B.17)

1  $N_{\text{houses}} \leftarrow 0$.

2  **for** $i_0 \in [M]$ **do**

3      $F(i_0) \leftarrow \{1, ..., k\}$.

4      **for** $j_1, \ldots, j_4$ *in* $\mathcal{N}^{k-1}_{i_0}$ *for which* $j_1 < j_4$ **do**

5         **if** $(i_0; j_1, j_2, j_3, j_4)$ *is a house* **then**

6            $N_{\text{houses}} \leftarrow N_{\text{houses}} + 1$.

7            **if** $N_{\text{houses}} \geq \Omega(k^{5k} \cdot M^5 \cdot n^{-4k+2})$ **then**

8               **return** None.

9         $\mathcal{I}' \leftarrow \{j_1, j_2, j_3, j_4\}$.

10         **if** $\bigcap_{a=1}^4 \mathcal{N}^{k-1}_{j_a} \backslash \{i_0\} \neq \emptyset$ **then**

11            Let $i_1$ be the (unique) element of $\bigcap_{a=1}^4 \mathcal{N}^{k-1}_{j_a} \backslash \{i_0\}$.

12            $\mathcal{I}'' \leftarrow \emptyset$.

13            $F(i_1) \leftarrow \{3, \cdots, k+2\}$.

14            **for** $z \in \{0, 1\}$ *and distinct* $\alpha, \beta, \gamma, \delta \in [4]$ *for which* $\alpha < \beta$ *and* $i_\gamma$ *(resp.* $i_\delta$*) is a $(k-1)$-neighbor of $i_\alpha$ (resp. $i_\beta$), and for which $i_0, \alpha, \beta$ are $(k-1)$-neighbors and $i_1, \gamma, \delta$ are $(k-1)$-neighbors* **do**

15               **if** *exactly $k-2$ choices of $i''$ which are $(k-1)$-neighbors of $i_z, i_\alpha, i_\beta$ and $(k-2)$-neighbors of $i_{1-z}, i_\gamma, i_\delta$, and which are not all $(k-1)$-neighbors of each other* **then**

16                  Add to $\mathcal{I}''$ all such $i''$.

17               **if** $|\mathcal{I}''| = 4k - 8$ **then**

18                  If $z = 0$, set $F(i_\alpha) \leftarrow \{1, 3, \ldots, k, k+1\}$, $F(i_\beta) \leftarrow \{2, 3, \ldots, k, k+1\}, F(i_\gamma) \leftarrow \{1, 3, \ldots, k, k+2\}$, and $F(i_\delta) \leftarrow \{2, 3, \ldots, k, k+2\}$.

19                  If $z = 1$, set $F(i_\alpha) \leftarrow \{1, 3, \ldots, k, k+1\}$, $F(i_\beta) \leftarrow \{1, 3, \ldots, k, k+2\}, F(i_{\gamma'}) \leftarrow \{2, 3, \ldots, k, k+1\}$ and $F(i_{\delta'}) \leftarrow \{2, 3, \ldots, k, k+2\}$.

20               **if** *exactly $\binom{k-2}{k-4}$ choices of $i'''$ which are $(k-2)$-neighbors of $i_0, i_1, i_\alpha, i_\beta, i_\gamma,$ and $i_\delta$, and which are also $(k-1)$-neighbors of at least one $i'' \in \mathcal{I}''$* **then**

21                  Let $\mathcal{I}'''$ denote the set of such $i'''$.

22                  $\mathcal{I} \leftarrow \{i_0, i_1\} \cup \mathcal{I}' \cup \mathcal{I}'' \cup \mathcal{I}'''$.

23                  Let $\mathbf{M}_{\text{sub}}$ denote the $\binom{k-2}{k-4} \times \binom{k-2}{k-4}$ submatrix of $\mathbf{M}$ given by restricting to the rows and columns indexed by $\mathcal{I}'''$ and subtracting 4 from every entry.

24                  $\_, G \leftarrow$ FINDFLORALSUBMATRIX($\mathbf{M}_{\text{sub}}, k - 2$).

25                  For every $i''' \in \mathcal{I}'''$, set $F(i''') \leftarrow G(i''') \cup \{1, 2, k+1, k+2\}$.

26                  **return** $\mathcal{I}, F$.

---

---

**Algorithm 4:** LEARNPRIVATEIMAGE($\{y^{\mathbf{X},w_i}\}_{i \in [m]}$)

---

**Input:** InstaHide dataset $\{y^{\mathbf{X},w_i}\}_{i \in [m]}$
**Output:** Vectors $\widetilde{x}_1, ..., \widetilde{x}_{k+2} \in \mathbb{R}^d$ equal to $k+2$ images (up to signs) from the original private dataset

1 $\mathbf{M} \leftarrow \frac{1}{k_{\mathsf{priv}}} \cdot$ GRAMEXTRACT($\{y^{\mathbf{X},w_i}\}, \frac{1}{2k_{\mathsf{pub}}+2k_{\mathsf{priv}}}$).

2 **for** $i \in [m]$ **do**

3  $\quad$ $S_i \leftarrow$ LEARNPUBLIC($\{([p_j]_S, y_j)\}_{j \in [d]}$).

4 **for** $i, j \in [m]$ **do**

5  $\quad$ $\mathbf{M}_{i,j} \leftarrow \mathbf{M}_{i,j} - \frac{1}{k_{\mathsf{pub}}} |S_i \cap S_j|$.

6 $\mathbf{M} \leftarrow k_{\mathsf{priv}} \cdot \mathbf{M}$.

7 $\mathcal{I}, F \leftarrow$ FINDFLORALSUBMATRIX($\mathbf{M}$).

8 For every pixel index $\ell \in [d]$, solve the system of equations $\left| \sum_{i \in F(j)} \widetilde{x}_i^{(\ell)} \right| = y_\ell^{\mathbf{X},w_j}$ in the variables $\{\widetilde{x}_i^{(\ell)}\}_{i \in [k_{\mathsf{priv}}+2]}$ for all $j \in \mathcal{I}$.

9 For every image index $i \in [k_{\mathsf{priv}}+2]$, let $\widetilde{x}_i \in \mathbb{R}^d$ denote the image whose $\ell$-th pixel is equal to $\widetilde{x}_i^{(\ell)}$.

10 **return** $\widetilde{x}_1, ..., \widetilde{x}_{k_{\mathsf{priv}}+2}$.

---

## C ADDITIONAL EXPERIMENTAL RESULTS

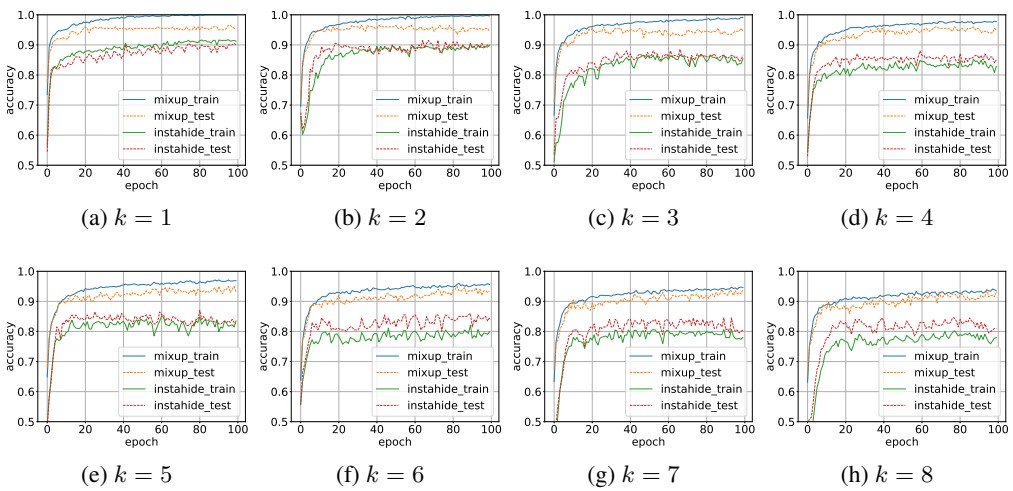

(a) $k = 1$ $\qquad$ (b) $k = 2$ $\qquad$ (c) $k = 3$ $\qquad$ (d) $k = 4$

(e) $k = 5$ $\qquad$ (f) $k = 6$ $\qquad$ (g) $k = 7$ $\qquad$ (h) $k = 8$

Figure 3: Comparing Vanilla, Mixup and Instahide training on Gaussian magnitude dataset with different $k$.

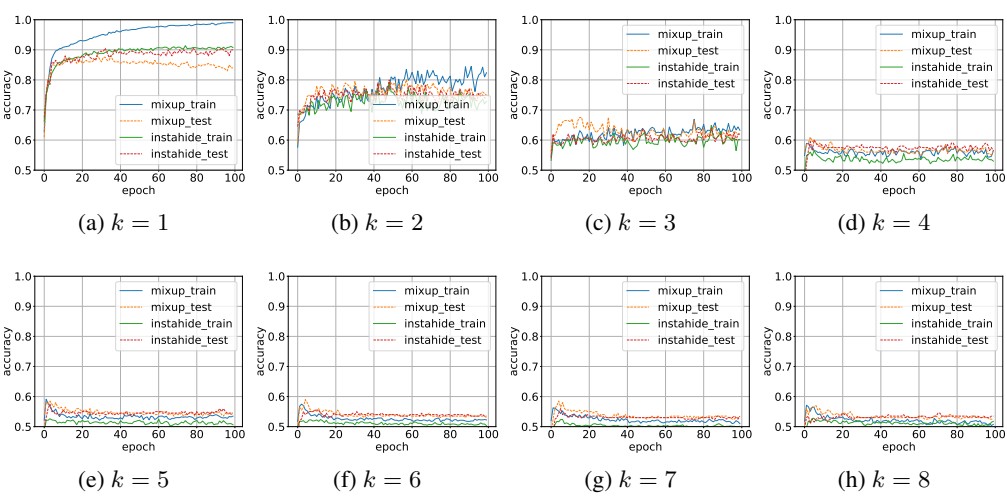

Figure 4: Comparing Vanilla, Mixup and Instahide training on Gaussian dataset with different $k$.

