# OpenReview forum: "On InstaHide, Phase Retrieval, and Sparse Matrix Factorization"
_ICLR.cc/2021/Conference — ICLR 2021 Poster_

### Official Review · AnonReviewer4 · 2020-10-29
**Answering the wrong question?**

**Rating:** 4
**Confidence:** 4

**Review:**

This paper investigates the security of InstaHide, a recently proposed algorithm for scrambling a secret image data set so that it is still useful for learning but doesn't allow inferences about individual images in the data set.

The paper provides an algorithm that takes the output of InstaHide, and reconstructs the secret images (in a particular parameter regime of very high-dimensional data). The attack works by reduction to a new variant of the phase retrieval problem with hidden inputs.

The paper's results appear to be correct. I am not familiar enough with the phase retrieval literature to judge the technical novelty and interest of the new problem variant and algorithm. The authors claim that the new variant is of independent interest, but don't really spend time justifying that.  I remain neutral on that point.

However, I disagree strongly with the authors' interpretation of their results for InstaHide's security, and do not think that they rise to the level of significance appropriate for ICLR.

To explain the setting a bit: InstaHide starts with two data sets of images (vectors in $\mathbb{R}^d$)  called the public and private data sets, each of size $n$. Given parameters $M$ and $k$, it produces $M$ synthetic output images. For each output image, one does the following:
1. select $k$ public images and $k$ private images (uniformly at random)
2. let $y$ be the sum of the selected images
3. Output a synthetic image $|y|$ consisting of the absolute values of the entries of $y$.

(I've simplified the scheme a bit—the $n$'s and $k$'s can be different for the public and private data—but the simplifications don't change this discussion.)

The paper mentions $k=2$ as a reasonable value.

The attack recovers the synthetic data set roughly when $d$ is reasonably large (at least $poly(k)\log(n)$) and when $M$, the number of synthetic data points, is very large (something like $n^k$). As mentioned above, the algorithm appears to be correct.

**On InstaHide's security**

I disagree with the authors' claims of their results' significance for security. I doubt that this paper would be acceptable at, say, a security or cryptography conference.

The main issue is that InstaHide looks broken to begin with, even when only 1 image is output.

The submission's main question is *"Under what settings can we rigorously prove that InstaHide preserves security of the private datasets used to generate the synthetic vectors?"*. Asking such a question requires first answering a seemingly much more basic question: *"What **clear, refutable conjecture** could one make about InstaHide's security that is both (a) sufficient to suggest it is reasonable to use on sensitive data and (b) not obviously false?*"


Unfortunately, I'm not sure there is such a conjecture. For InstaHide to be a reasonable approach to image data privacy, it should be hold up to settings where an attacker knows quite a bit about the format of the input. Suppose, for example, that the images themselves are "sparse" in the following sense: each consists of small grayscale image, randomly placed on a much larger black background. Then the linear combination of a small number of them would also be sparse with high probability, and its absolute values would exactly reveal all of the component grayscale images. One would start getting attacks with $M=1$. For a more realistic but messy example, consider a setting in which the images are screenshots where most of each screen is text but  sensitive pictures may appear in different parts of the screen. Finally, Matthew Jagielski shared the following notebook with many people (including the authors of InstaHide) which works out a simple example with 2 images:
https://colab.research.google.com/drive/1ONVjStz2m3BdKCE16axVHZ00hcwdivH2?usp=sharing
(The images, one of a dog and the other of a bird, are easy to pick out from the absolute values of the combination.)

**Implications for this submission**

Because of the known problems with InstaHide,  it's not clear to me what light the submission really sheds on InstaHide's security. The example attacks above shows that it isn't hard to find natural distributions and settings of $k$ for which InstaHide allows recovery of the inputs. (Aside: doesn't the requirement of $M\approx n^k$ makes this attack unlikely to be practical, unless very small $k$ is necessary for utility?) The question in my mind is whether any reasonably robust security or privacy claim could be made. The last twenty years have seen major advances in how to formulate such claims, but those advances are not reflected in either the original paper or this follow-up.

I would encourage the authors to rethink the framing for the question they address. Perhaps they can argue that the complexity of their setting is important for understanding InstaHide's security, or perhaps they can make a case that the new phase retrieval problem and algorithm are interesting in their own right (and rewrite the paper as a function of that)? Absent such a reframing of the paper, I don't think it should be accepted.

**Minor Comments**

* Page 2:  "to ensure we are not working in an uninteresting setting where InstaHide has zero utility, we empirically verify that in the setting of Theorem 1.1, one can train on the synthetic vectors and get good test accuracy on the original Gaussian dataset". It is fine to do experiments in the setting where InstaHide is useful, but ensuring high utility would seem to me to make attacks easier, not harder.

* Page 3: "consider the extreme case where InstaHide only works with private feature vectors[...], so that the only information we have access to is the synthetic vector generated by InstaHide. In this case, it is clearly information-theoretically impossible to recover anything about [which combinations were chosen] or the private dataset." I strongly disagree. I don't see why that's true in the Gaussian setting—isn't the mutual information between the synthetic data and the secret data infinite (since the coefficient vectors are discrete)? In any case, the "sparse images" setting above gives a simple illustration of how absolute values of linear combinations can leak tons of information.

**Follow-up to author comments**

I appreciate the authors' thoughtful response. While I agree on certain points, I am not convinced that the paper, as currently written, is ready for acceptance to a venue like ICLR. (That said, I also still believe the paper's core technical claims to be correct; my question is about significance.)

The paper's premise is that reconstruction attacks on InstaHide deserve extensive investigation. I'd like to see justification for that premise. To be absolutely clear: I strongly believe that techniques (like InstaHide) which attempt to provide security against moderately strong attackers deserve discussion and investigation. However, I also believe that the starting point for such work should be a careful attempt to formulate security goals. I don't see how the current paper advances the important parts of that discussion. The focus on reconstruction is narrow; my score reflects that.

To add just a little bit to my review: InstaHide is best viewed as a proposed lightweight alternative to multiparty computation (MPC). MPC protocols allow participants to compute on shared data in a way that reveals nothing but the final outcome of the computation (tools like FHE, to which InstaHide's authors compare it, can help achieve that goal but are not qualitatively different).

MPC protocols (and InstaHide in particular) say nothing about how much is revealed about the data by the final trained model (the "ideal functionality", in the language of MPC). There is at this point a large literature showing that models themselves leak information in surprising ways (membership inference, to pick an example that received recent attention).

Even if we focus on "lightweight MPC" as the end goal, the literature on data privacy suggests a wide range of more sophisticated measures of security than resistance/vulnerability to reconstruction. (That is, it's interesting and potentially important to relax the goal of full simulation that one normally aims for in MPC; but then one should spell out what the relaxed goal is, why it's sufficient for some settings, etc.) This submission reflects none of the past decades' lessons on that count.

Responding to specific points:

* I was not comparing to the paper of Carlini et al. I assume that this submission and the manuscript of Carlini et al are independent.

* (Minor point) I'll stick by my complaint about Page 3: "it is clearly information-theoretically impossible to recover anything about [...] the private dataset." I understand and agree with the math of the rebuttal, but not with the conclusion. To spell out my original objection: Let $V=(V_1,V_2)$ be the random variable consisting of the two private images, and let $W$ be their average. It is not true that the mutual information $I(V;W)$ is 0 (which is the natural meaning of "no information about the data set"). It is not even true that $I(V_1;W)$ is 0.  Concretely, learning $W$ makes certain pairs of images much more likely than they were a priori. Whether that's ok depends on the context, what else is likely to be  known about the images, etc. My point isn't that one released image will lead to a practical attack; it is that information leaks in lots of ways. If you want to claim that leaked information isn't useful for an attack, that requires a clear notion of "useful for an attack" (and possibly a proof, though that's less important than a clear claim).

---

> ### Author Response · Authors · 2020-11-20
> **Thanks for the review (1/2)**
>
> Thanks for taking the time to write this review.
> 1. We actually seem to be on the same side about a lot of these issues. For one, we strongly agree that there are plenty of natural settings where using InstaHide would be highly problematic.
>     - The interpretation we give for our result, namely the paragraph right before Section 1.1, is actually rather conservative. We say “we must be careful about the properties we posit about the underlying distribution generating the public and private data.” The reviewer’s “sparse image” and “text+sensitive image” examples and Jagielski’s attack appear to reinforce this point.
> 2. To paraphrase, the reviewer’s main criticism is: people were already aware of these examples that suggest “InstaHide looks broken to begin with, even when only 1 image is output,” so what does this submission bring to the conversation? **The answer**: *our work gives the first algorithmic techniques for attacking InstaHide in a natural setting where one cannot simply “eyeball” a single synthetic image after absolute value post-processing and deduce nontrivial information about the private images.*
>     - A simple setting of this flavor would be a distribution where the features are largely independent of each other and similarly distributed, of which the Gaussian setting we work with is a clean abstraction. Not only does the intuition behind the abovementioned examples tell us nothing about this setting, but **as the reviewer’s misunderstanding in the second bullet point in “Minor Comments” demonstrates, this intuition can even be misleading:**
>     - Given just a single InstaHide image generated from a private Gaussian dataset, as we state on P. 3, it is impossible to recover a private image. To elaborate, if a synthetic vector v is given by the entrywise absolute value of 0.5v_1 + 0.5v_2 for vectors v_1,v_2~N(0,Id), then an equally plausible pair of vectors generating v would be w_1,w_2 given by swapping the i-th entry of v_1 with that of v_2 for any collection of indices i in {1,...,d}. Any of these (exponentially many in d) such pairs is equally likely under the Gaussian measure.
>     - In light of known hardness results for various inverse problems over Gaussian space (e.g. continuous LWE, statistical query lower bounds for various learning problems), it is not unreasonable to hope that the recovery problem (Problem 1 on P. 4), or some version which allows some error in some reasonable norm, is hard for Gaussian datasets, perhaps under a plausible average-case complexity assumption. Admittedly, this need not guarantee more nuanced notions of privacy, but this is clearly a mathematically well-posed, refutable conjecture. While such a claim is insufficient on its own to justify use of InstaHide, it would be a step towards formalizing a security claim wherein InstaHide is safe to use for *certain kinds of datasets*.
>     - We show that an ambitious version of such a conjecture is false: for Gaussian datasets, InstaHide is not secure against adversaries that can afford to run in time/samples $n_{priv}^{O(k_{priv})}$. More optimistically, this leaves open whether there is an $n_{priv}^{\Omega(k_{priv})}$ lower bound (keep in mind that k_priv can be as large as 6 or 8 without a significant loss in utility).
>     - Finally, note that the Gaussian setting we study falls exactly under the kind of threat model suggested by the reviewer, namely where “the attacker knows quite a bit about the format of the input”: we are literally telling the attacker the exact distribution that the data comes from.
> 3. The title of the review is “Answering the wrong question?” Indeed, is the theory we develop here relevant for anything security-wise? The answer is yes, based on subsequent developments:
>     - A month after our work and shortly after this review was posted, Carlini et al. independently gave a heuristic attack which was able to recover all n_priv = 100 private images using the 5000 synthetic images provided in the InstaHide challenge, which took k_priv = 2 and k_pub = 4.
>     - Crucially, the main sub-problem they give a heuristic for is to extract information from a similarity matrix, which is precisely the Gram matrix in our work, but specialized to k_priv = 2. This exact algorithmic question is the focus of Sections A.3 onwards in our paper.
>     - While our work is focused on the Gaussian setting, the floral submatrix-based procedure we give for our sub-problem provably works for any dataset for which the Gram matrix can be constructed, and could just as well have been used in Carlini et al.’s attack to reconstruct some of the private images (as we discuss below, our runtime guarantee is quite practical in this regime).
>     - We reiterate that our algorithm has rigorous guarantees for all k_priv, while in contrast, Carlini et al.’s attack, while highly impressive, is heuristic and has so far only been demonstrated to work empirically when k_priv = 2.

---

> > ### Author Response · Authors · 2020-11-20
> > **Thanks for the review (2/2)**
> >
> > Other comments:
> > 1. Regarding Jagielski’s attack, note that we say on P. 1 that k_pub = k_priv = 2 is a reasonable choice, not k_priv + k_pub = 2. Indeed, it has been empirically observed that Jagielski’s attack fails if you mix more than just two images, so this example doesn’t say much in the setting of our paper.
> > 2. **Runtime considerations**: The reviewer wrote: “(Aside: doesn't the requirement of M≈n^k make this attack unlikely to be practical, unless very small k is necessary for utility?)”
> >     - The parameter regime to keep in mind is k_pub = k_priv = 2 or 3, as this is the setting under which Huang et al. demonstrated one can get utility. For these choices, we only require n_priv^{4/3} and n_priv^{5/2} synthetic images respectively (see Theorem A.1). Note that when n_priv = 100 and k_priv = 2, as in the InstaHide challenge, n^{4/3} < 500, whereas the InstaHide challenge released 5000 synthetic images. Our runtime is then dominated by the time it takes to form the 500x500 Gram matrix.
> >     - We also point out that the attack for learning the public coordinates (Step 1 in the Proof Overview in Section 3) is quite practical as it only requires roughly O(k^2 log(n)) synthetic feature vectors and runs in time poly(n,k). Step 2 is also practical as it just involves entrywise applying a simple univariate function to every entry of an empirical covariance matrix, and Step 2 alone already reveals important information, e.g. whether two selection vectors used the same private image.
> > 3. The reviewer wrote: “It is fine to do experiments in the setting where InstaHide is useful, but ensuring high utility would seem to me to make attacks easier, not harder.”
> >     - Our experiments are for a very specific choice of labeling function (see “Settings” in Section 4.2), so they should be interpreted as *nonzero*, rather than high, utility. We don’t believe that being able to learn this particular function says anything about how difficult it is to attack InstaHide over Gaussians.

---

### Official Review · AnonReviewer1 · 2020-11-02
**Review for "What Can Phase Retrieval Tell Us About Private Distributed Learning?"**

**Rating:** 8
**Confidence:** 4

**Review:**

Summary: The paper examines the security of a recently proposed privacy scheme, InstaHide Huang et al. 2020, that can be used to generate synthetic training data. Under a standard Gaussian distributional assumption on the data, the authors propose an algorithm that can extract private information from the synthetic vectors generated by InstaHide.

Comments:

InstaHide is a recently proposed technique Huang et al. 2020, that aggregates local data into synthetic data that can both preserve the privacy of the local datasets and be used to train good models. InstaHide assumes presence of both public feature vectors (e.g. a publicly available dataset like ImageNet) and a collection of private feature vectors (which we want to hide). This scheme is based on an interesting looking idea, and also seems to achieve good empirical performance (in that by training a network like ResNet on the generated synthetic data one can achieve good test accuracy on the real data). This paper investigates the security properties of the InstaHide scheme. The main result shows that if the public and private data are both drawn from an i.i.d. Gaussian distribution then it is possible to reconstruct (some of) the private data if we have the access to output of the InstaHide on multiple queries and public data. The reconstruction is correct up to sign of the coordinates..

This is an interesting result, implying that a careful discussion of security for InstaHide should be sensitive to the properties of the distributions generating the private and public feature vectors.

The paper is well-presented, however, most of the interesting theoretical results are in the appendices. The idea behind the reconstruction can be broken into four steps: a) identify the public coordinates of the selection vector, b) construct the Gram matrix of the selection vectors, c) identify a particular submatrix of the Gram matrix, and d)  use this particular submatrix to identify private feature vectors.

I have not verified all the technical details, but overall the reconstruction idea looks sound. The theoretical results are also backed by some simple proof-of-concept experiments.

Some questions:
1)	Would have nice to see the formal statement of the primary theorem in the main paper. Also what is the overall probability of success in Theorem A.1?
2)	Can be the Gaussian assumption on the X matrix be relaxed to a subgaussian assumption.

Side-remark: I thought the title of the paper is slightly misleading. The result is specific to InstaHide scheme and is not a general statement about private distributed learning.

---

> ### Author Response · Authors · 2020-11-20
> **Thanks for the positive feedback!**
>
> Thanks so much for taking the time to read and understand our paper, and thanks for the positive comments and the great questions! We agree that the title should be changed and intend to rename it to "On InstaHide, Phase Retrieval, and Sparse Matrix Factorization." In response to the two questions:
> 1. We will update the draft to include the theorem statement in the intro. As for the success probability, it can be 1 - delta for arbitrary constant delta > 0, and we just pay a corresponding extra constant term in the runtime/sample complexity. In case it's of interest, we elaborate a bit here on the quantitative dependence:
>     - Our algorithm works as long as the private/public data and the random selection vectors satisfy the following deterministic conditions 1) the matrices $\tilde{M}$ in Algorithm 1 and $\hat{\Sigma}$ in Algorithm 2 concentrate sufficiently in max-norm and entrywise respectively, and 2) the conditions of Lemmas A. 14 and A.16, namely that the underlying hypergraph is sparse and there aren't too many "houses," hold.
>     - By standard concentration, condition 1) holds with probability 1 - delta as long as the number of private/public images scales with log(1/delta)
>     - Condition 2) holds with probability 1 - delta as long as the number of selection vectors scales with poly(1/delta) (concentration is weaker for 2 because we're simply using second moment method).
> 2. Certain parts of our algorithm can indeed be relaxed to the subgaussian case.
>     - Our algorithms in A.3 onwards (which constitute the bulk of the technical heavy lifting in this work) for recovery using the Gram matrix apply to any distribution over public/private images for which the Gram matrix can be constructed.
>     - It is probably possible to extend our phase retrieval-based techniques for recovering the public coordinates to the sub-Gaussian case. The main place there where we used exact Gaussianity was when we used Stein's lemma in the proof of Lemma A.3. There have been some works giving algorithms for phase retrieval with sub-Gaussian measurement vectors (see e.g. an analysis of PhaseLift in this setting: https://arxiv.org/pdf/1604.07281.pdf), and it seems very plausible that these can be extended to the missing-entry setting that is relevant to InstaHide.
>     - Whether our procedure for constructing the Gram matrix, which uses the closed-form relationship between the covariance of a Gaussian and its folded counterpart, can be extended to the subgaussian case is an interesting question for future study.

---

### Official Review · AnonReviewer2 · 2020-11-05
**Comments on private distributed learning**

**Rating:** 7
**Confidence:** 3

**Review:**

In this paper, the security of private datasets is considered. Huang et al. (2020) proposed InstaHide for this problem. The InstaHide can be summarised as follows:

From a random convex combination of k_pub public and k_private vectors,
Multiply every coordinate of the resulting vector by an independent random sign in {+-1},and define this to be the synthetic feature vectors.
In the paper, the authors try to understand InstaHide by phase retrieval. The main result of this paper is to show when the private and public data is Gaussian, then they can recover the private feature vector form the given synthetic dataset and the public feature vectors

I think the topic is good. Under many situations, we should focus on security of private information. But in practice, the public and private dataset can be any. The assumptions in this paper may be not correct. The problem of recovery of private vector can indeed be considered as a phase retrieval problem from public vector and the synthetic dataset generated from InstalHide. I think it is the affine phase retrieval or phase retrieval with background information.

---

> ### Author Response · Authors · 2020-11-20
> **Thanks for your positive feedback!**
>
> Thanks so much for the positive comments! It is encouraging to see that you agree this is a worthwhile topic to pursue.

---

### Official Review · AnonReviewer3 · 2020-11-06
**Provably working attack to Instahide algorithm**

**Rating:** 7
**Confidence:** 3

**Review:**

The purpose of the paper seems clear: it proposes an attack to the recently proposed algorithm called Instahide (ICML 2020) which is a probabilistic algorithm for generating synthetic private data in the distributed setting. The attack proposed in this paper is considered for the case where the private data is i.i.d. Gaussian distributed, and Thm 1.1 says that one can recover k original feature vectors with O(k^2) + O(M^2) computational complexity, where M is the total number of original data elements.

Although the original paper describing Instahide proposes few attacks and has some analysis for them, and revolves around the probabilistic notions of privacy, it lacks any rigorous privacy guarantees for the hiding algorithm (e.g. differential privacy (DP) guarantees) and the illustration of the performance of Instahide seems to be only heuristic.

This paper gives an attack which underlines the fact that theoretical guarantees of the kind given by DP are indeed very important. As there is an attack for the Gaussian case (as given here), the assumptions on the distributions of the original data have to be something different if theoretical guarantees for Instahide are desired.

The experiments with small dense neural networks are mainly to illustrate the that in the setting of the assumptions used is not useless in a sense that it is possible to train a neural network to perform classification with data satisfying the assumptions.

 I think the paper is very well written, and, although I didn't read in detail all the proofs given of Appendix, those I read seem to be correct and I have no reason to suspect there are errors.

My only slight concern is whether some more theoretically oriented venue would suit better for this paper than ICLR, however the topic is very relevant to ICLR.

---

> ### Author Response · Authors · 2020-11-20
> **Thanks for your positive feedback!**
>
> We completely agree that the issue of understanding what can be done in the space of private distributed learning is quite timely and important to this venue. We also agree that the focus of our paper skews more to the theoretical side of things. Our motivation for this is that there has been a lot of discussion about the security of InstaHide, and in our opinion the right way to approach such discussions seems to be to adopt a mathematically rigorous formulation for what exactly people are claiming InstaHide can/cannot do. Anyways, thanks so much for taking the time to read and understand our results, and thanks for the positive comments!

---

### Author Response · Authors · 2020-11-23
**Updated draft: some revisions to presentation**

We updated our draft to incorporate the helpful comments from all of the reviewers. The primary changes were in presentation:
- Changing the title to more accurately reflect the nature of our contribution
- Including (an informal version of) the main theorem statement in the introduction
- Elaborating on why the Gaussian setting is important for understanding the security of InstaHide, in particular how it is a useful and challenging testbed for understanding security of InstaHide on datasets where features are at most weakly dependent and similarly distributed (unlike natural images)
- A discussion of Jagielski's attack as well as a detailed comparison (see Appendix A in the supplement) between our contributions and that of the very recent work of Carlini et al. which appeared subsequent to this submission. In Appendix A, we highlight an important commonality between our techniques and theirs, namely extracting information from the Gram matrix by implicitly solving a row-sparse Boolean matrix factorization problem, and describe in detail how our approaches to this problem relate to/differ from each other.
- Instead of deferring proofs in the appendix to a later section, we have integrated these proofs into the main flow of the argument to improve the clarity of presentation. The mathematical content remains exactly the same.
- The only (minor) mathematical change is that we noticed a slight error in the stated runtime of the original formal theorem statement which we have since corrected.

---

### Decision · Program_Chairs · 2021-01-07
**Final Decision**

**Decision:**

Accept (Poster)

**Comment:**

The paper considers an attack of the recently proposed InstaHide algorithm mixing up public and private images by convex combination to achieve security of sensitive data. The paper formulates the problem as a multi-task phase retrieval problem with missing-data, and shows that under Gaussian data distribution setting, we can recover a small number of private data samples given sufficiently large dimensionality and number of synthetic samples output by InstaHide.

Theoretically, the Gaussian data distribution is quite restrictive in practice, but it could be a good start. The paper also uses some novel techniques in the analysis, which meets the technical standard of ICLR. The reviewer mainly concerns about the general motivation and formulation of "security" studied in the paper, since attacks can be trivial in practical scenarios where data is non-Gaussian, which reveals a possible weakness on practical value of this work.


Although the work is probably better suited for a theoretical-oriented conference, I nevertheless feel it should be also acceptable for ICLR because it specifically addresses a recent distributed learning problem and the results are non-trivial and improving our understanding of the InstaHide's security.